# Uni3DETR: Unified 3D Detection Transformer

**Zhenyu Wang**[1]   **Yali Li**[1] *   **Xi Chen**[2]   **Hengshuang Zhao**[2] *   **Shengjin Wang**[1]

[1] Department of Electronic Engineering, Tsinghua University, BNRist
[2] The University of Hong Kong
{wangzy20@mails., liyali13@, wgsgj@}tsinghua.edu.cn, {xchen2, hszhao}@cs.hku.hk

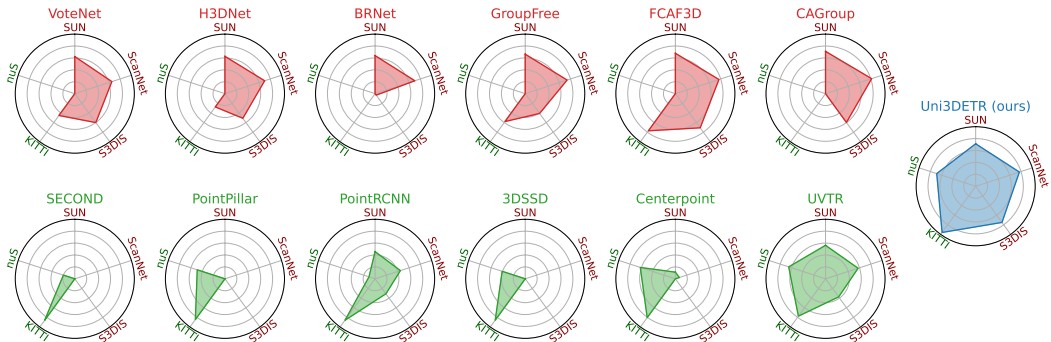

Figure 1: The high-level overview comparing the performance of 12 existing 3D detectors and our Uni3DETR on a broad range of 3D object detection datasets: indoor datasets SUN RGB-D (SUN), ScanNet, S3DIS, and outdoor datasets KITTI, nuScenes (nuS). The metrics are $AP_{25}$ for indoor datasets, $AP_{70}$ on moderate difficulty car for KITTI, and NDS for nuScenes. The center of the circle means that the corresponding metric is less than 10%, and the outermost means 90%. Existing indoor detectors are plotted in red and outdoor detectors are in green. Our model has the remarkable capacity to generalize across a wide range of diverse 3D scenes (a larger polygon area).

## Abstract

Existing point cloud based 3D detectors are designed for the particular scene, either indoor or outdoor ones. Because of the substantial differences in object distribution and point density within point clouds collected from various environments, coupled with the intricate nature of 3D metrics, there is still a lack of a unified network architecture that can accommodate diverse scenes. In this paper, we propose **Uni3DETR**, a unified 3D detector that addresses indoor and outdoor 3D detection within the *same* framework. Specifically, we employ the detection transformer with point-voxel interaction for object prediction, which leverages voxel features and points for cross-attention and behaves resistant to the discrepancies from data. We then propose the mixture of query points, which sufficiently exploits global information for dense small-range indoor scenes and local information for large-range sparse outdoor ones. Furthermore, our proposed decoupled IoU provides an easy-to-optimize training target for localization by disentangling the $xy$ and $z$ space. Extensive experiments validate that Uni3DETR exhibits excellent performance consistently on both indoor and outdoor 3D detection. In contrast to previous specialized detectors, which may perform well on some particular datasets but suffer a substantial degradation on different scenes, Uni3DETR demonstrates the strong generalization ability under heterogeneous conditions (Fig. 1). Codes are available at https://github.com/zhenyuw16/Uni3DETR.

---

*Corresponding author

37th Conference on Neural Information Processing Systems (NeurIPS 2023).

# 1 Introduction

3D object detection from point clouds aims to predict the oriented 3D bounding boxes and the semantic labels for the real scenes given a point set. Unlike mature 2D detectors [41, 19, 56, 4] on RGB images, which have demonstrated the ability to effectively address diverse conditions, the problem of 3D object detection has been considered under different scenarios, leading to the development of distinct benchmarks and methodologies for each. Specifically, 3D detection approaches are currently addressed separately for indoor and outdoor scenes.

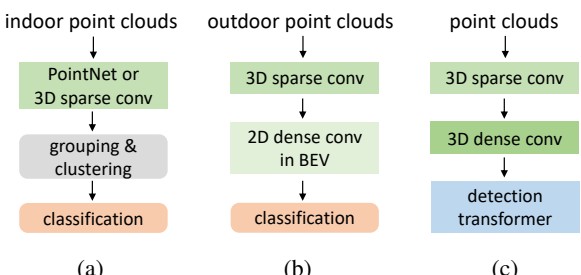

Figure 2: **Illustration of the structures of previous indoor 3D detectors (a), outdoor detectors (b) and our model.** Our model has the capacity for 3D detection in both indoor and outdoor scenes.

Indoor 3D detectors [38, 70, 62, 58] usually adopt the "grouping, clustering and classification" manner (Fig. 2a), while outdoor detectors [64, 24, 52, 49] typically convert the features into the 2D bird's-eye-view (BEV) space (Fig. 2b). Although these two tasks only differ in their respective application contexts, the optimal approaches addressing them exhibit significant differences.

The key challenge in developing a unified 3D detector lies in the substantial disparities of point cloud data collected in indoor and outdoor environments. In general, indoor scenarios are cluttered, where various objects are *close and dense*, occupying the majority of the scene. In comparison, objects in outdoor scenes are *small and sparse*, where background points dominate the collected point clouds. Such disparities result in the lack of a unified 3D detector: 1) For the detection head, because of the severe distraction of excessive background points in outdoor point clouds, together with the hyperparameter sensitivity associated with grouping, the grouping-based indoor detectors are infeasible for outdoor 3D detection. Besides, outdoor objects are separated clearly and not overlapped in the BEV space, which does not apply to indoor objects. The height overlap among indoor objects makes the manner of detecting in the BEV space inappropriate for indoor detection. 2) For the feature extractor, existing backbones in 3D detectors are similarly designed for a singular scene. Indoor detectors are usually equipped with point-based [39, 40] or 3D sparse convolution based [10, 43] backbone, where point-based models are usually susceptible to the diverse structures of points under different scenes, and sparse convolution models are deficient in representing the features of object centers. In contrast, the 2D convolutions in outdoor detectors for extracting BEV features easily lead to information loss for indoor detection.

In this paper, we propose a Unified 3D DEtection TRansformer (**Uni3DETR**) based on only point clouds for detecting in diverse environments (Fig. 2c). Two attributes of our Uni3DETR contribute to its universality for both indoor and outdoor scenes. First, we employ a hybrid structure that combines 3D sparse convolution and dense convolution for feature extraction. The pure 3D architecture avoids excessive height compression for indoor point clouds. Simultaneously, the sparse convolution prevents the huge memory consumption for large-range outdoor data, and the dense convolution alleviates the center feature missing problem for sparse outdoor points. Second, we utilize transformer [57] for 3D object detection. The set-to-set prediction way in transformer-based detectors [4, 77] directly considers predicted objects and ground-truth labels, thus tends to be resistant to the distinction from data themselves. The transformer decoder is built on the extracted voxel feature and we formulate queries as 3D points from the scene. The points and voxels interact through cross-attention, which well adapts to the characteristics of 3D data.

Based on our 3D detection transformer, we further propose two necessary components for universality under various scenes. One is the mixture of query points. Specifically, besides the learnable query, we introduce the non-learnable query initialized by sampling the original points and the voxelized points, and integrate the learnable and non-learnable queries for feeding the transformer. We observe that the learnable query points mostly contain local information and fit the outdoor detection well, while non-learnable query points emphasize global information thus are more effective for dense indoor scenes. The other one is the decoupled 3D IoU. Compared to the usual 3D IoU, we decouple the $x, y$ and $z$ axis in the decoupled IoU to provide stronger positional regularization, which not only involves all directions in the 3D space but also is beneficial for optimization in transformer decoders.

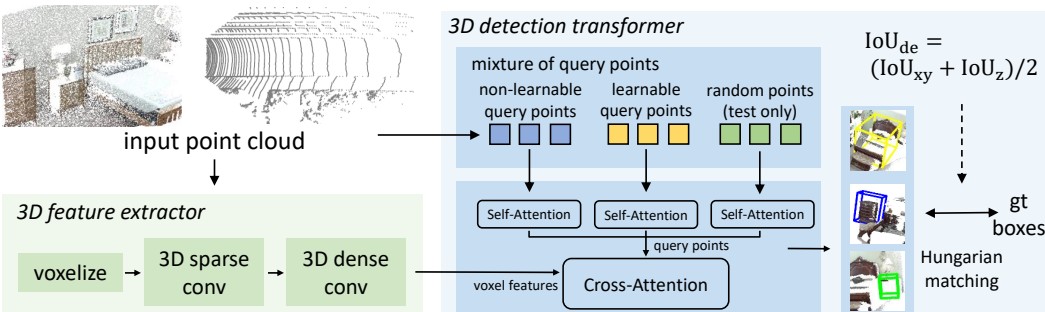

Figure 3: **The overall architecture of our Uni3DETR.** We use the hybrid combination of 3D sparse convolution and dense convolution for 3D feature extraction, and our detection transformer for set prediction. The mixture of query points contributes to the sufficient usage of global and local information, and the decoupled IoU provides more effective supervision about localization.

Our main contributions can be summarized as follows:

- We propose Uni3DETR, a unified 3D detection transformer that is designed to operate effectively in both indoor and outdoor scenes. To the best of our knowledge, this is the first point cloud based 3D detector that demonstrates robust generalization across diverse environments.
- We propose the mixture of query points that collaboratively leverages the learnable and non-learnable query. By aggregating local and global information, our mixture of queries fully explores the multi-scene detection capacity.
- We decouple the multiple directions in the 3D space and present the decoupled 3D IoU to bring more effective positional supervision for the transformer decoder.

Our Uni3DETR achieves the state-of-the-art results on both indoor [53, 12, 1] and outdoor [16, 3] datasets. It obtains the 67.0% and 71.7% AP$_{25}$ on the challenging SUN RGB-D and ScanNet V2 indoor datasets, and the 86.7% AP in the moderate car category on KITTI validation. On the challenging nuScenes dataset, Uni3DETR also achieves the 61.7% mAP and 68.5% NDS.

## 2 Uni3DETR

### 2.1 Overview

We present the overall architecture of our Uni3DETR in Fig. 3. It consists of a 3D feature extractor and the detection transformer for detecting in various 3D scenes. The mixture of query points is fed into transformer decoders to predict 3D boxes, under the supervision of decoupled 3D IoU.

**3D feature extractor.** Existing 3D detectors usually adopt point-based [40] or voxel-based [64, 43] backbones for feature extraction. Considering that point-based backbones are vulnerable to the specific structure of point clouds themselves and less efficient in point set abstraction, we utilize the voxel-based model for extracting 3D features. After voxelization, we utilize a series of 3D sparse convolution layers to encode and downsample 3D features, to avoid the overload memory consumption for large-range outdoor scenes. Then, we convert the extracted sparse features into the dense ones and apply 3D dense convolutions for further feature processing. The dense convolution alleviates the feature missing problem of center points.

### 2.2 Unified 3D Detection Transformer for Diverse Scenes

**Detection transformer with point-voxel interaction.** We employ the transformer structure based on the extracted voxel features and the set prediction manner for 3D detection. Motivated by recent 2D transformer-based detectors [29, 25, 69] that formulate queries as anchor boxes, we regard 3D points in the 3D space as queries. Its structure is illustrated in Fig. 4.

Specifically, we denote $P_q = (x_q, y_q, z_q)$ as the $q$-th point to represent the $q$-th object, $C_q \in \mathbb{R}^D$ is its content query, where $D$ is the dimension of decoder embeddings. We introduce the deformable

attention [77] for the cross-attention module and view $P_q$ as the reference point for point-voxel interaction. Unlike the deformable DETR decoder, here we directly learn the reference point $P_q$. Suppose the voxel feature as $V$, the process of the cross-attention is modeled as:

$$\text{CrossAtt}(C_q, V) = \text{DeformAtt}(C_q + \text{MLP}(\text{PE}(P_q)), V, P_q) \tag{1}$$

where PE denotes the sinusoidal positional encoding. The transformer decoder predicts the relative positions for the query points: $(\Delta x_q, \Delta y_q, \Delta z_q)$. Based on the relative predictions, the query points are refined layer-by-layer.

Compared to 2D detection transformers, because of the stronger positional information in the 3D space, our detection transformer requires less layers for query refinement. To further utilize the information across multiple layers, we average predictions from all transformer decoders except for the first one.

**Mixture of query points.** In the learning process, the above query points gradually turn around the object instances for better 3D detection. As they are updated during training, we refer them as *learnable query points*. Since such learnable query points finally concentrate on the points near the objects, they primarily capture the local information of the scene. For outdoor point clouds, where objects are sparse and small within the large-range scene, the learnable query points help the detector avoid the disturbance of the dominated background points.

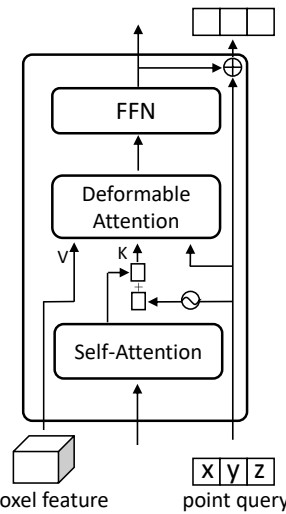

In comparison, for the indoor scene, the range of 3D points is significantly less and the objects in relation to the entire scene are of comparable size. In this situation, besides the local information, the transformer decoder should also utilize the global information to consider the whole scene. We thus introduce the *non-learnable query points* for global information. Specifically, we use the Farthest Point Sampling (FPS) [40] for the input point cloud data and take the sampled points as the query points. These query points are frozen during training. Because of the FPS, the sampled points tend to cover the whole scene and can well compensate for objects that learnable query points ignore.

Figure 4: **The structure of our detection transformer.** The decoder is built on extracted voxel features and we introduce 3D points as the query. The points and voxels interact through cross-attention.

As we utilize the voxel-based backbone for feature extraction, voxelization is a necessary step for processing the point cloud data. After voxelization, the coordinates and numbers of the raw point clouds may change slightly. To consider the structure of both the raw data and points after voxelization, what we use is two kinds of non-learnable query points $P_{nl}$ and $P_{nlv}$. $P_{nl}$ denotes the non-learnable query points from FPS the original point clouds, and $P_{nlv}$ comes from sampling the points after voxelization. Together with the learnable query points $P_l$, the mixture of query points ultimately consists of three ones: $P = \{P_l, P_{nl}, P_{nlv}\}$. We follow [5] to perform group-wise self-attention in the transformer decoders for the mixture of query points, where these three kinds of query points do not interact with each other. After transformer decoders, three sets of predicted 3D boxes $\{bb_l, bb_{nl}, bb_{nlv}\}$ are predicted respectively. We apply one-to-one assignment to each set of 3D boxes independently using the Hungarian matching algorithm [23] and calculate the corresponding training loss to supervise the learning of the detection transformer.

At the test time, besides the above three kinds of query points, we further generate a series of additional points uniformly and randomly in the 3D voxel space as another set of query points $P_{rd}$. These randomly generated points evenly fill the whole scene, thus well compensating for some potentially missed objects, especially for the indoor scenes. As a result, four groups of query points $P = \{P_l, P_{nl}, P_{nlv}, P_{rd}\}$ are utilized for inference. Ultimately, four sets of predicted 3D boxes $\{bb_l, bb_{nl}, bb_{nlv}, bb_{rd}\}$ are produced at the test time by Uni3DETR. Because of the one-to-one assignment strategy, the individual box sets do not require additional post-processing methods for removing duplicated boxes. However, there are still overlapped predictions among these four sets. We thus conduct box merging finally among the box sets to fully utilize the global and local information and eliminate duplicated predictions. Specifically, we cluster the 3D boxes based on the 3D IoU among them and take the median of them for the final prediction. The confidence score of the final box is the maximum one of the clustered boxes.

## 2.3 Decoupled IoU

Existing transformer-based 2D detectors usually adopt the L1 loss and the generalized IoU loss [42] for predicting the bounding box. The generalized IoU here mitigates the issue of L1 loss for different scales thus provides further positional supervision. However, for 3D bounding boxes, the calculation of IoU becomes a more challenging problem. One 3D box is usually denoted by: $bb = (x, y, z, w, l, h, \theta)$, where $(x, y, z)$ is the center, $(w, l, h)$ is the size and $\theta$ is the rotation. The usual 3D IoU [72] between two 3D boxes $bb_1$ and $bb_2$ is calculated by:

$$\text{IoU}_{3D} = \frac{\text{Area}_{\text{overlapped}} \cdot z_{\text{overlapped}}}{\text{Area}_1 \cdot z_1 + \text{Area}_2 \cdot z_2 - \text{Area}_{\text{overlapped}} \cdot z_{\text{overlapped}}} \tag{2}$$

where $\text{Area}$ denotes the area of rotated boxes in the $xy$ space.

From Equ. 2, we notice that the calculation of 3D IoU generally consists of two spaces: the $xy$ space for the bounding box area, involving the $x, y, w, l, \theta$ variables, and the $z$ space for the box height. The two spaces are multiplied together in the 3D IoU. When supervising the network, the gradients of these two spaces are coupled together - optimizing in one space will interfere with another, making the training process unstable. For 3D GIoU [72], besides the negative coupling effect, the calculation of the smallest convex hull area is even non-optimizable. As a result, the usual 3D IoU is hard to optimize for detectors. For our detection transformer, where L1 loss already exists for bounding box regression, the optimization problem of 3D IoU severely diminishes its effect in training, thus being hard to mitigate the scale problem of L1 loss.

Therefore, an ideal metric for transformer decoders in 3D should at least meet the following demands: First, it should be easy to optimize. Especially for the coupling effect for different directions, its negative effect should be alleviated. Second, all shape properties of the 3D box need to be considered, so that accurate supervision signals can be provided for all variables. Third, the metric should be scale invariant to alleviate the issue of L1 loss. Motivated by these, we forward the decoupled IoU:

$$\text{IoU}_{de} = \left(\frac{\text{Area}_{\text{overlapped}}}{\text{Area}_1 + \text{Area}_2 - \text{Area}_{\text{overlapped}}} + \frac{z_{\text{overlapped}}}{z_1 + z_2 - z_{\text{overlapped}}}\right)/2 \tag{3}$$

Specifically, our decoupled IoU is the average of the IoU in the $xy$ space and the $z$ space. Under the summation operation, the gradients from the two items remain separate and independent, avoiding any coupling effect. According to previous research [47], the coupling issue does not exist in 2D or 1D IoU. As a result, the decoupled IoU effectively circumvents the negative effects of coupling. The scale-invariant property of 3D IoU is also well-reserved for the decoupled IoU. These make our decoupled IoU well suit the training of the transformer decoder.

The decoupled IoU is introduced in both the training loss and the matching cost. We also introduce it into classification and adopt the variant of quality focal loss [26]. Denote the binary target class label as $c$, the predicted class probability as $\hat{p}$, the classification loss for Uni3DETR is:

$$L_{cls} = -\hat{\alpha}_t \cdot |c \cdot \text{IoU}_{de} - \hat{p}|^\gamma \cdot \log(|1 - c - \hat{p}|) \tag{4}$$

where $\hat{\alpha}_t = \alpha \cdot c \cdot \text{IoU}_{de} + (1 - \alpha) \cdot (1 - c \cdot \text{IoU}_{de})$. The above classification loss can be viewed as using the soft target $\text{IoU}_{de}$ in focal loss [28]. The decoupled IoU is easy to optimize, allowing us to avoid the need for performing the stop-gradient strategy on $\text{IoU}_{de}$. Therefore, the learning of classification and localization is not separated, assisting Uni3DETR predicting more accurate confidence scores. Equ. 4 will force the predicted class probability $\hat{p}$ towards $\text{IoU}_{de}$, which might be inconsistent with $\text{IoU}_{3D}$ used in evaluation. We thus follow [22] and introduce an IoU-branch in our network to predict the 3D IoU. The normal $\text{IoU}_{3D}$ supervises the learning of the IoU-branch and the binary cross-entropy loss is adopted for supervision. The weighted geometric mean of the predicted 3D IoU and the classification score is utilized for the final confidence score. The final loss function for training Uni3DETR is thus the classification loss (Equ. 4), the L1 loss and the IoU loss with $\text{IoU}_{de}$ for bounding box regression, and the binary cross-entropy loss for IoU prediction.

## 3 Experiments

To demonstrate the universality of our Uni3DETR under various scenes, we conduct extensive experiments in this section. We evaluate Uni3DETR in the indoor and outdoor scenes separately.

Table 1: **The performance of Uni3DETR for indoor 3D object detection.** The main comparison is based on the best results of multiple experiments. We re-implement VoteNet, H3DNet, and GroupFree on the S3DIS dataset. * indicates the multi-modal method with both point clouds and RGB images.

| Method | SUN RGB-D | | ScanNet | | S3DIS | |
|---|---|---|---|---|---|---|
| | $AP_{25}$ | $AP_{50}$ | $AP_{25}$ | $AP_{50}$ | $AP_{25}$ | $AP_{50}$ |
| ImVoteNet* [37] | 63.4 | - | - | - | - | - |
| TokenFusion* [60] | 64.9 | 48.3 | 70.8 | 54.2 | - | - |
| VoteNet [38] | 57.7 | 35.7 | 58.6 | 33.5 | 58.0 | 25.3 |
| GSDN [17] | - | - | 62.8 | 34.8 | 47.8 | 25.1 |
| H3DNet [70] | 60.1 | 39.0 | 67.2 | 48.1 | 50.9 | 22.0 |
| BRNet [9] | 61.1 | 43.7 | 66.1 | 50.9 | - | - |
| 3DETR [35] | 59.1 | 32.7 | 65.0 | 47.0 | - | - |
| VENet [62] | 62.5 | 39.2 | 67.7 | - | - | - |
| GroupFree [30] | 63.0 | 45.2 | 69.1 | 52.8 | 42.8 | 19.3 |
| RBGNet [59] | 64.1 | 47.2 | 70.6 | 55.2 | - | - |
| HyperDet3D [71] | 63.5 | 47.3 | 70.9 | 57.2 | - | - |
| FCAF3D [43] | 64.2 | 48.9 | 71.5 | 57.3 | 66.7 | 45.9 |
| **Uni3DETR (ours)** | **67.0** | **50.3** | **71.7** | **58.3** | **70.1** | **48.0** |

**Datasets.** For indoor 3D detection, we evaluate Uni3DETR on three indoor 3D scene datasets: SUN RGB-D [53], ScanNet V2 [12] and S3DIS [1]. SUN RGB-D is a single-view indoor dataset with 5,285 training and 5,050 validation scenes, annotated with 10 classes and oriented 3D bounding boxes. ScanNet V2 contains 1,201 reconstructed training scans and 312 validation scans, with 18 object categories for axis-aligned bounding boxes. S3DIS consists of 3D scans from 6 buildings, 5 object classes annotated with axis-aligned bounding boxes. We use the official split, evaluate our method on 68 rooms from Area 5 and use the rest 204 samples for training. We use the mean average precision (mAP) under IoU thresholds of 0.25 and 0.5 for evaluating on these three datasets.

For outdoor 3D detection, we conduct experiments on two popular outdoor benchmarks: KITTI [16] and nuScenes [3]. The KITTI dataset consists of 7,481 LiDAR samples for its official training set, and we split it into 3,712 training samples and 3,769 validation samples for training and evaluation. The nuScenes dataset is a large-scale benchmark for autonomous driving, using the 32 lanes LiDAR for data collection. We train on the 28,130 frames of samples in the training set and evaluate on the 6,010 validation samples. We use mAP and nuScenes detection score (NDS), a weighted average of mAP and other box attributes like translation, scale, orientation, velocity.

**Implementation details.** We implement Uni3DETR with mmdetection3D [11], and train it with the AdamW [32] optimizer. We set the number of learnable query points to 300 for datasets except for nuScenes, where we set to 900. For indoor datasets, we choose the 0.02m grid size. For the KITTI dataset, we use a (0.05m, 0.05m, 0.1m) voxel size and for the nuScenes, we use the (0.075m, 0.075m, 0.2m) voxel size. The nuScenes model is trained with 20 epochs, with the CBGS [75] strategy. For outdoor datasets, we also conduct the ground-truth sampling augmentation [64] and we remove the ground-truth sampling at the last 4 epochs. Dynamic voxelization [73] and ground-truth repeating [21] are also adopted during training. Besides these data-related parameters, other architecture-related hyper-parameters are all the same for different datasets.

### 3.1 Indoor 3D Object Detection

We first train and evaluate Uni3DETR on indoor 3D detection datasets and list the comparison with existing state-of-the-art indoor 3D detectors in Tab. 1. Here we omit some grouping methods like [58], which relies on mask annotations for better grouping and clustering. Our method obtains the 67.0% $AP_{25}$ and 50.3% $AP_{50}$ on SUN RGB-D, which surpasses FCAF3D, the state-of-the-art indoor detector based on the CNN architecture, by almost 3%. On the ScanNet dataset, Uni3DETR surpasses FCAF3D by 1% on $AP_{50}$. It is also noticeable that with only point clouds participating in training, our method even obtains better performance than existing multi-modal approaches that require both point clouds and RGB images. On the SUN RGB-D dataset, our model is 2.1% higher on $AP_{25}$ than TokenFusion. This strongly demonstrates the effectiveness of Uni3DETR. The visualized results of Uni3DETR on SUN RGB-D can be seen in the left two of Fig. 5.

Our method also significantly outperforms existing transformer-based indoor detectors, 3DETR and GroupFree. The superiority of our method is more significant, especially in localization: Uni3DETR

Table 2: **The performance of Uni3DETR for outdoor 3D object detection on the KITTI validation set with 11 recall positions.** $M$: ✓ means training on three classes and the blank means training only on the car class. *: AP on the moderate car is the most important metric.

| Method | $M$ | Car-3D (IoU=0.7) | | | Ped.-3D (IoU=0.5) | | | Cyc.-3D (IoU=0.5) | | |
|---|---|---|---|---|---|---|---|---|---|---|
| | | Easy | Mod.* | Hard | Easy | Mod. | Hard | Easy | Mod. | Hard |
| SECOND [64] | ✓ | 88.61 | 78.62 | 77.22 | 56.55 | 52.98 | 47.73 | 80.59 | 67.16 | 63.11 |
| PointPillar [24] | ✓ | 86.46 | 77.28 | 74.65 | 57.75 | 52.29 | 47.91 | 80.06 | 62.69 | 59.71 |
| PointRCNN [51] | ✓ | 89.06 | 78.74 | 78.09 | 67.69 | 60.74 | 55.83 | 86.16 | 71.16 | 67.92 |
| Part-$A^2$ [52] | ✓ | 89.56 | 79.41 | 78.84 | 65.69 | 60.05 | 55.45 | 85.50 | 69.90 | 65.49 |
| PV-RCNN [49] | ✓ | 89.35 | 83.69 | 78.70 | 63.12 | 54.84 | 51.78 | 86.06 | 69.48 | 64.50 |
| CT3D [46] | ✓ | 89.11 | 85.04 | 78.76 | 64.23 | 59.84 | 55.76 | 85.04 | 71.71 | 68.05 |
| RDIoU [47] | ✓ | 89.16 | 85.24 | 78.41 | 63.26 | 57.47 | 52.53 | 83.32 | 68.39 | 63.63 |
| **Uni3DETR (ours)** | ✓ | **89.61** | **86.57** | **78.96** | **70.18** | **62.49** | **58.32** | **87.18** | **72.90** | **68.86** |
| 3DSSD [65] | | 88.82 | 78.58 | 77.47 | - | - | - | - | - | - |
| STD [66] | | 89.70 | 79.80 | 79.30 | - | - | - | - | - | - |
| Voxel-RCNN [13] | | 89.41 | 84.52 | 78.93 | - | - | - | - | - | - |
| VoTr-TSD [33] | | 89.04 | 84.04 | 78.68 | - | - | - | - | - | - |
| CT3D [46] | | 89.54 | 86.06 | 78.99 | - | - | - | - | - | - |
| BtcDet [63] | | - | 86.57 | - | - | - | - | - | - | - |
| RDIoU [47] | | 89.76 | 86.62 | 79.04 | - | - | - | - | - | - |
| **Uni3DETR (ours)** | | **90.23** | **86.74** | **79.31** | - | - | - | - | - | - |

Table 3: **The performance of Uni3DETR for outdoor 3D object detection on the nuScenes validation set.** We compare with previous methods without test-time augmentation. *: the implementation from OpenPCDet [55].

| Method | NDS(%) | mAP(%) | mATE↓ | mASE↓ | mAOE↓ | mAVE↓ | mAAE↓ |
|---|---|---|---|---|---|---|---|
| PointPillar [24] | 49.1 | 34.3 | 0.424 | 0.284 | 0.529 | 0.377 | 0.194 |
| CBGS [75] | 61.5 | 51.9 | - | - | - | - | - |
| CenterPoint [67] | 64.9 | 56.6 | 0.291 | 0.252 | 0.324 | 0.284 | 0.189 |
| VoxelNeXt* [8] | 66.7 | 60.5 | 0.301 | 0.252 | 0.406 | 0.217 | 0.186 |
| PillarNet [48] | 67.4 | 59.8 | 0.277 | 0.252 | 0.289 | 0.247 | 0.191 |
| UVTR [27] | 67.7 | 60.9 | 0.334 | 0.257 | 0.300 | 0.204 | 0.182 |
| **Uni3DETR (ours)** | **68.5** | **61.7** | 0.288 | 0.249 | 0.303 | 0.216 | 0.181 |

outperforms them by 5.1% on SUN RGB-D and 5.5% on ScanNet in $AP_{50}$. Such results validate that compared with existing transformers in 3D detection, our detection transformer on voxel features with the mixture of query points is more appropriate for 3D detection.

## 3.2 Outdoor 3D Object Detection

**KITTI.** We then conduct experiments on the outdoor KITTI dataset. We report the detection results on the KITTI validation set in three difficulty levels - easy, moderate, and hard in Tab. 2. We notice that Uni3DETR also achieves the satisfying performance with the *same* structure as that for indoor scenes. For the most important KITTI metric, AP on the moderate level of car, we obtain the 86.57% AP, which is more than 1.5% higher than CT3D and 1.3% higher than RDIoU. With only the car class in training, the car moderate AP is 86.74%, which is also higher than existing methods like BtcDet. Its ability in outdoor scenes is thus demonstrated. The superiority of Uni3DETR is also consistent for the pedestrian and cyclist class. This illustrates that our model can also distinguish small and scarce objects well, which is one main aspect that hinders existing indoor detectors in the outdoor environments. The visualized results are shown in the right two of Fig. 5.

**nuScenes.** We further conduct experiments on the nuScenes dataset. Compared to the KITTI dataset, the range of scenes in nuScenes is larger, with 360 degrees around the LiDAR instead of only the front view. The point cloud in nuScenes is also more sparse (with 32-beam LiDAR compared to the KITTI 64 beams). These make the nuScenes more challenging and even some existing outdoor detectors fail to address the detection problem on nuScenes. We list the comparative results in Tab. 3. We obtain the 68.5% NDS, which surpasses recent methods like PillarNet, UVTR, VoxelNeXt. Compared to the most recent method VoxelNeXt, Uni3DETR is 1.8% higher in NDS and 1.4% higher in mAP. Besides the detection metric mAP, Uni3DETR also achieves promising results for predicting other attributes of boxes. The ability of our model in the outdoor scenes is further validated.

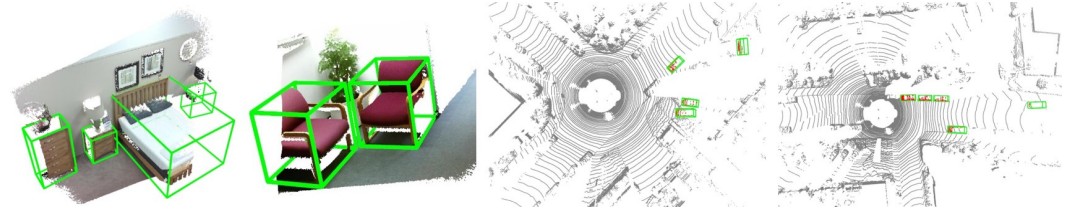

Figure 5: **The visualized results of Uni3DETR** on the indoor SUN RGB-D dataset (the left two) and the outdoor KITTI dataset (the right two). Better zoom in with colors.

Table 4: **The 3D performance of our detection transformer** on the SUN RGB-D dataset compared with some existing transformer decoder structures.

| transformer | $AP_{25}$ | $AP_{50}$ |
|---|---|---|
| 3DETR [35] | 59.1 | 32.7 |
| Deform [77] | 61.3 | 45.4 |
| DAB-Deform [29] | 62.0 | 44.3 |
| DN-Deform [25] | 62.0 | 44.6 |
| ours ($\{P_l\}$) | 62.6 | 46.4 |
| ours (mixed) | 66.4 | 49.6 |

Table 5: **Effect of the mixture of query points on 3D indoor and outdoor detection.** We compare with different combinations of queries. *: AP on the moderate car is the most important metric for KITTI.

| query | SUN RGB-D | | KITTI car | | |
|---|---|---|---|---|---|
| | $AP_{25}$ | $AP_{50}$ | Easy | Mod.* | Hard |
| $\{P_l\}$ | 62.6 | 46.4 | 90.20 | 85.59 | 78.89 |
| $\{P_{nl}\}$ | 54.0 | 39.9 | 89.42 | 79.24 | 78.29 |
| $\{P_{nlv}\}$ | 57.4 | 43.1 | 89.70 | 79.34 | 78.14 |
| $\{P_l, P_{nl}\}$ | 65.1 | 46.9 | 90.06 | 85.94 | 78.64 |
| $\{P_l, P_{nlv}\}$ | 64.5 | 46.9 | 90.04 | 85.86 | 78.75 |
| $\{P_l, P_{nl}, P_{nlv}\}$ | 66.4 | 49.6 | 90.12 | 86.26 | 79.01 |
| $\{P_l, P_{nl}, P_{nlv}, P_{rd}\}$ | 67.0 | 50.3 | 90.23 | 86.74 | 79.31 |

## 3.3 Ablation Study

**3D detection transformer.** We first compare the detection AP of our detection transformer with existing transformer structures and list the results in Tab. 4. As 3DETR is built on point-based features, its performance is restricted by the less effectiveness of point representations. We then implement other methods based on voxel features. As anchors are negatively affected by the center point missing in 3D point clouds, formulating queries as 3D anchors like DAB- or DN-DETR is less effective than our query points. In comparison, our detection transformer better adapts the 3D points, thus achieves the highest performance among existing methods. Besides, it accommodates the mixture of query points, which brings further detection improvement.

**Mixture of query points.** Tab. 5 analyzes the effect of the mixture of query points on 3D object detection. For indoor detection on SUN RGB-D, the simple usage of the learnable query points $\{P_l\}$ obtains the 62.6% $AP_{25}$. Substituting the learnable query with the non-learnable query hurts the performance. This is because non-learnable query points are fixed during training and cannot adaptively attend to information around objects. After we introduce $\{P_l, P_{nl}\}$, we find such mixed usage notably boosts $AP_{25}$, from 62.6% to 65.1%: the global information in $P_{nl}$ overcomes the lack of local information in $P_l$ and contributes to more comprehensive detection results. As a result, the object-related metric is enhanced greatly. Utilizing $\{P_l, P_{nlv}\}$ similarly improves $AP_{25}$, but less than the effect of $P_{nl}$. This is because voxelization may remove some points, making the global information not that exhaustive. With $\{P_l, P_{nl}, P_{nlv}\}$, the three different query points further enhance the performance. $P_{nlv}$ here provides more location-accurate global information, thus $AP_{50}$ is improved by almost 3%. Random points at the test time further brings a 0.6% improvement.

For outdoor detection on KITTI, we notice that the effect of non-learnable query points is less significant than that in indoor scenes. Using $\{P_l, P_{nl}\}$ is just 0.35% higher than $\{P_l\}$. Since outdoor point clouds are large-range and sparse, the effect of global information in non-learnable query will be counteracted by excessive background information. Therefore, local information matters more in the outdoor task. The collaboration of the two-level information in the mixture of query makes it suitable for 3D detection in various scenes.

We further combine multiple groups of learnable query points and list the performance in Tab. 6 to compare with the mixture of query points. Such a combination is actually analogous to Group DETR [5, 6] in 2D detection. We observe that unlike 2D detection, multiple groups of learnable queries do not bring an improvement. Even though the number of groups has been raised to 7, the AP on SUN RGB-D still keeps almost the same. The reason is that multiple groups in 2D mainly speed up the convergence, which does not matter much in 3D. In this situation, by providing multi-level information about the scene, the mixture of query points works better for 3D detection.

Table 6: **Comparison with multiple groups of learnable query points** on the SUN RGB-D dataset.

| query | $AP_{25}$ | $AP_{50}$ |
|---|---|---|
| $\{P_l\}$ | 62.6 | 46.4 |
| $\{P_l\} \times 2$ | 62.2 | 46.2 |
| $\{P_l\} \times 3$ | 62.7 | 46.6 |
| $\{P_l\} \times 5$ | 62.3 | 46.7 |
| $\{P_l\} \times 7$ | 62.7 | 46.6 |
| $\{P_l, P_{nl}\}$ | 65.1 | 46.9 |
| $\{P_l, P_{nl}, P_{nlv}\}$ | 66.4 | 49.6 |

Table 7: **Effect of the decoupled IoU on 3D indoor and outdoor detection.** $IoU_{aa-3D}$ denotes the 3D IoU in axis-aligned way, $IoU_{xy}$ is the 2D IoU in the $xy$ space, $IoU_z$ is the 1D IoU in the $z$ space.

| IoU | SUN RGB-D | | KITTI car | | |
|---|---|---|---|---|---|
| | $AP_{25}$ | $AP_{50}$ | Easy | Mod.* | Hard |
| w/o IoU | 20.7 | 3.7 | 69.81 | 61.97 | 64.85 |
| $IoU_{3D}$ | 48.3 | 29.4 | 88.70 | 78.52 | 77.60 |
| $IoU_{aa-3D}$ | 50.4 | 31.6 | 89.57 | 79.44 | 76.81 |
| RD-IoU [47] | 29.1 | 14.0 | 82.08 | 73.50 | 73.98 |
| $IoU_{xy}$ | 65.8 | 49.0 | 89.91 | 79.58 | 77.93 |
| $IoU_z$ | 64.7 | 44.1 | 90.09 | 79.74 | 78.57 |
| $IoU_{de}$ | 67.0 | 50.3 | 90.23 | 86.74 | 79.31 |

Table 8: **Comparison of performance and computational complexity against existing methods on the indoor SUN RGB-D and the outdoor nuScenes dataset.** The metrics are $AP_{25}$ for SUN RGB-D and mAP for nuScenes. The computational cost is measured on a single RTX 3090 GPU.

| Method | performance | | | efficiency | | |
|---|---|---|---|---|---|---|
| | avg. | indoor | outdoor | latency | params | FLOPS |
| CenterPoint [67] | 37.75 | 18.9 | 56.6 | 0.32 s | 9.17 M | 121.10 G |
| VoxelNeXt [8] | 39.30 | 18.1 | 60.5 | 0.29 s | 7.12 M | 42.57 G |
| PillarNet [48] | 44.00 | 28.2 | 59.8 | 0.31 s | 12.55 M | 100.10 G |
| UVTR [27] | 55.55 | 50.2 | 60.9 | 0.51 s | 26.12 M | 451.12 G |
| **Uni3DETR (ours)** | 64.35 | 67.0 | 61.7 | 0.52 s | 26.71 M | 458.74 G |

**Decoupled IoU.** In Tab. 7, we compare our decoupled IoU with different types of IoU in the 3D space. On the SUN RGB-D dataset, optimizing with no IoU loss, only classification and L1 regression loss, just obtains the 20.7% $AP_{25}$, indicating that 3D IoU is necessary to address the scale problem of L1 loss. Introducing the normal $IoU_{3D}$ alleviates the limitation of L1 loss to some extent, boosting the $AP_{25}$ to 48.3%. However, it is restricted by the negative coupling effect, thus is even inferior to axis-aligned 3D IoU, which does not take the rotation angle into consideration. In comparison, since this problem does not exist in 2D IoU, even the $IoU_{xy}$, without considering the $z$ axis, can be 17.5% higher than $IoU_{3D}$. Our decoupled IoU, which can be viewed as $(IoU_{xy} + IoU_z)/2$, takes all directions into account, thus further improves the $AP_{25}$ to 67.0%. On the outdoor KITTI dataset, the decoupled IoU is equally critical, 24.77% higher than w/o IoU and 8.22% higher than $IoU_{3D}$. Its necessity for transformer-based 3D detection is further validated.

### 3.4 Comparison of Computational Cost

Here we further list the comparison of both performance and efficiency in the Tab. 8. We can observe that the computational budget compared with these methods is not significant: the inference time (latency) is almost the same as UVTR and the FLOPS is only 1.16% more than UVTR. In addition, we obtain significantly better detection performance on both indoor and outdoor datasets. Compared with VoxelNeXt, one model that mainly focuses on reducing the FLOPS of 3D detectors, we achieve more than 40% indoor AP and more than 25% average AP improvement. In this paper, we mainly target a unified structure. To ensure that the detector can accommodate both indoor and outdoor detection, we have, to a certain extent, made sacrifices in terms of efficiency, in order to prioritize its unified ability. A more efficient and unified structure can be left as one of the future works.

### 3.5 Discussion

**A unified voxel size.** The word "unified" in our paper specifically refers to the architecture aspect. Since point clouds are collected with different sensors, their ranges and distributions vary significantly (about 3m for indoor but more than 70m for outdoor datasets). For the experiments above, we adopt different values for the voxel size, one data-related parameter. We further conduct the experiment with the same KITTI voxel size (0.05m, 0.05m, 0.1m) and list the results in Tab. 9. Compared with other outdoor detectors, our superiority is still obvious. Therefore, even if standardizing such a data-related parameter, our model can still obtain a higher AP.

Table 9: **Comparison with existing methods** on the indoor SUN RGB-D dataset and outdoor KITTI car dataset with the same voxel size. *: results from training with different voxel sizes.

| Method | indoor | outdoor |
|---|---|---|
| 3DSSD [65] | 9.5 | 78.6 |
| CenterPoint [67] | 18.9 | 74.4 |
| UVTR [27] | 35.9 | 72.0 |
| Uni3DETR (ours) | **47.3** | **86.7** |
| Uni3DETR (ours) * | **67.0** | **86.7** |

Table 10: **The cross-dataset performance on the indoor SUN RGB-D dataset** compared with existing methods. We compare with the RGB image based Cude RNN with its metric $AP_{3D}$.

| Method | Trained on | $AP_{3D}$ |
|---|---|---|
| Cube RCNN [2] | SUN RGB-D | 34.7 |
| Cube RCNN | $OMNI3D_{IN}$ | 35.4 |
| Uni3DETR (ours) | SUN RGB-D | 64.3 |
| Uni3DETR (ours) | ScanNet | 50.9 |

**Cross-dataset experiments.** We further conduct the cross-dataset evaluation with the Omni3D metric [2]. From the results in Tab. 10, it is worth noticing that Uni3DETR has a good cross-dataset generalization ability. The cross-dataset AP (ScanNet to SUN RGB-D) is 16.2% higher than Cube RCNN trained on the SUN RGB-D dataset. The reason is that our Uni3DETR takes point clouds as input for 3D detection, while Cube RCNN takes RGB images for detection. By introducing 3D space information from point clouds, the superiority of a unified architecture for point clouds over Cube RCNN can be demonstrated. We further emphasize that cross-dataset evaluation is a more difficult problem for point cloud based 3D object detection, as the dataset-interference issue is more serious. We believe our Uni3DETR can become the basic platform and facilitate related research.

**Limitation.** One limitation is that we still require separate models for different datasets. Currently, some methods [2, 68] have tried one single model for multiple datasets for 3D object detection. We hope our Uni3DETR can become the foundation for indoor and outdoor 3D dataset joint training.

## 4 Related Work

Existing research on **3D Object Detection** has been separated into indoor and outdoor categories. Indoor 3D detectors [38, 70, 9, 58] usually cluster points first based on extracted point-wise features, then conduct classification and detection. Recently, 3D sparse convolution based detectors [43, 45] also adopt the anchor-free manner [56] in 2D for 3D indoor detection. In comparison, outdoor 3D detectors [64, 52, 49, 13, 46, 7, 50] usually convert 3D features into the 2D BEV space and adopt 2D convolutions for predicting 3D boxes. However, because of the significant difference in point clouds between indoor and outdoor scenes, a unified 3D detector in various environments is still absent. Some recent methods [20, 31, 60, 44] have performed experiments on both indoor and outdoor datasets. However, they rely on RGB images to bridge the difference of point clouds.

**Transformer-based object detectors** have been widely used in 2D object detection [4, 77, 34, 29, 25, 69], predicting object instances in the set-to-set manner. Recently, the transformer structure has been introduced in 3D object detection. Methods like [33, 18, 15, 54, 14] introduce transformers into the backbone for 3D feature extraction, while still adopting CNNs for predicting boxes. In comparison, approaches like [35, 61, 36, 74, 76] leverage transformers in the detection head for box prediction. However, these methods still rely on point-wise features or BEV features for either indoor or outdoor 3D detection, thus are restricted by the singular scene.

## 5 Conclusion

In this paper, we propose Uni3DETR, a unified transformer-based 3D detection framework that addresses indoor and outdoor 3D object detection within the same network structure. By feeding the mixture of query points into the detection transformer for point-voxel interaction and supervising the transformer decoder with the decoupled IoU, our Uni3DETR fills the gap of existing research in unified 3D detection under various scenes. Extensive experiments demonstrate that our model can address both indoor and outdoor 3D detection with a unified structure. We believe our work will stimulate following research along the unified and universal 3D object detection direction, and Uni3DETR will become a basic tool and platform.

**Acknowledgment.** This work is supported by the National Key Research and Development Program of China in the 14th Five-Year with Nos. 2021YFF0602103 and 2021YFF0602102, National Natural Science Foundation of China (No. 62201484), HKU Startup Fund and Seed Fund for Basic Research.

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

# A  Datasets and Implementation Details

**SUN RGB-D** [53]. When training on the SUN RGB-D dataset, the input point clouds are filtered in the range [-3.2m, 3.2m] for the $x$ axis, [-0.2m, 6.2m] for the $y$ axis and [-2m, 0.56m] for the $z$ axis. The grid size is set to 0.02m. We randomly flip the input data along the $x$ axis and randomly sample 20,000 points for data augmentation. The common global translation, rotation and scaling strategies are also adopted here. We train Uni3DETR with the initial learning rate of 1.67e-4 and the batch size of 32 for 90 epochs, and the learning rate is decayed by 10x on the 70th and 80th epoch. The ratio of the classification score and the predicted IoU is 0.8:0.2.

**ScanNet** [12]. For the ScanNet dataset, here we adopt the range of [-6.4m, 6.4m] for the $x$ and $y$ axis and [-0.1m, 2.46m] for the $z$ axis after global alignment, with the 0.02m grid size. Here we do not adopt the RGB information of the dataset for training. The input data are randomly flipped along both the $x$ and $y$ axis. We utilize dynamic voxelization [73] on the ScanNet dataset. The initial learning rate is set to 7.5e-5, with the batch size of 24. We train Uni3DETR for 240 epochs, where the learning rate is decayed on the 192nd and the 228th epoch. Other hyper-parameters and operations are the same as the SUN RGB-D dataset.

**S3DIS** [1]. As the S3DIS dataset point clouds are distributed only on the positive planes, which is adverse to the random flipping augmentation, we first translate the input data to make it centered at the original point. Besides the coordinate information, we also utilize the RGB information of the point clouds. We train the Uni3DETR for 520 epochs, with the learning rate decaying at the 416th and 494th epoch. The batch size is set to 8, with the initial learning rate of 3.33e-5. Other hyper-parameters and operations are the same as the ScanNet dataset.

**KITTI** [16]. For the KITTI dataset, the data augmentation operations are basically the same as previous methods like [13]. For the ground-truth sampling augmentation, we sample at most 20 cars, 10 pedestrians and 10 cyclists from the database. 18000 points are randomly sampled at the training time. During training, we also adopt van class objects as car objects. The ground-truth sampling augmentation and the object-level noise strategy are removed at the last 2 epochs. We train Uni3DETR for 70 epochs, with the learning rate decaying at the 64th epoch. The initial learning rate is set to 9e-5, with the batch size of 8. As the KITTI dataset suffers from the sparse objects seriously, we follow [21] to repeat the ground truth labels 5 times during training. The predicted car objects are filtered at the threshold of 0.5 after inference. The ratio of the classification score and the predicted IoU is 0.2:0.8 on the KITTI dataset.

**nuScenes** [3]. Compared to the KITTI dataset, the nuScenes dataset covers a larger range, with 360 degrees around the LiDAR instead of only the front view. The point cloud in nuScenes is also more sparse (with 32-beam LiDAR compared to the KITTI 64 beams). Besides, the nuScenes dataset contains 10 classes, with the severe long-tail problem. The initial learning rate is set to 1e-5, with the batch size of 16 and the cyclic schedule. We train the Uni3DETR for 20 epochs.

# B  Test Set Results

We further conduct the experiment and evaluate our method on the test set of KITTI and nuScenes dataset. The comparison is listed in the Tab. 11 and Tab. 12 separately. For the most important KITTI metric, AP on the moderate level of car, we obtain the 82.26% AP, which is 0.83 points higher than PV-RCNN, 0.49 points higher than CT3D, and 0.38 points higher than PV-RCNN++. On the test set of the nuScenes dataset, we obtain the 65.2% mAP and 70.8% NDS. The consistent superiority further demonstrates the ability of Uni3DETR on outdoor 3D detection.

# C  Visualized Results

**Comparison results about the mixture of query points.** We first provide comparative visualized results in Fig. 6 to illustrate the effectiveness of the mixture of query points. For the first case, it can be seen that training with only the learnable query points concentrates on the right region of the bed and the nightstand, and ignores the left sofa. This similarly applies to the rest two cases. For the second case, the right nightstand is not detected and for the third case, three chairs are ignored. The common point among these three cases is that these ignored objects are partly occluded, thus with insufficient points. The limited quantities of point clouds therefore restrict the performance of the 3D

Table 11: **The performance of Uni3DETR for outdoor 3D object detection on the KITTI test set with 40 recall positions.** We train the models on the car category only. *: AP on the moderate car is the most important metric.

| Method | Easy | Mod.* | Hard |
|---|---|---|---|
| SECOND [64] | 88.61 | 78.62 | 77.22 |
| PointPillar [24] | 82.58 | 74.31 | 68.99 |
| Part-$A^2$ [52] | 87.81 | 78.49 | 73.51 |
| PV-RCNN [49] | 90.25 | 81.43 | 76.82 |
| CT3D [46] | 87.83 | 81.77 | 77.16 |
| PV-RCNN++ [50] | - | 81.88 | - |
| **Uni3DETR (ours)** | **91.14** | **82.26** | **77.58** |

Table 12: **The performance of Uni3DETR for outdoor 3D object detection on the nuScenes test set.** We compare with previous methods with double-flip testing.

| Method | mAP(%) | NDS(%) |
|---|---|---|
| PointPillar [24] | 30.5 | 45.3 |
| CBGS [75] | 52.8 | 63.3 |
| CenterPoint [67] | 58.0 | 65.5 |
| Focals Conv [7] | 63.8 | 70.0 |
| UVTR [27] | 63.9 | 69.7 |
| PillarNet [48] | 65.0 | 70.8 |
| **Uni3DETR (ours)** | **65.2** | **70.8** |

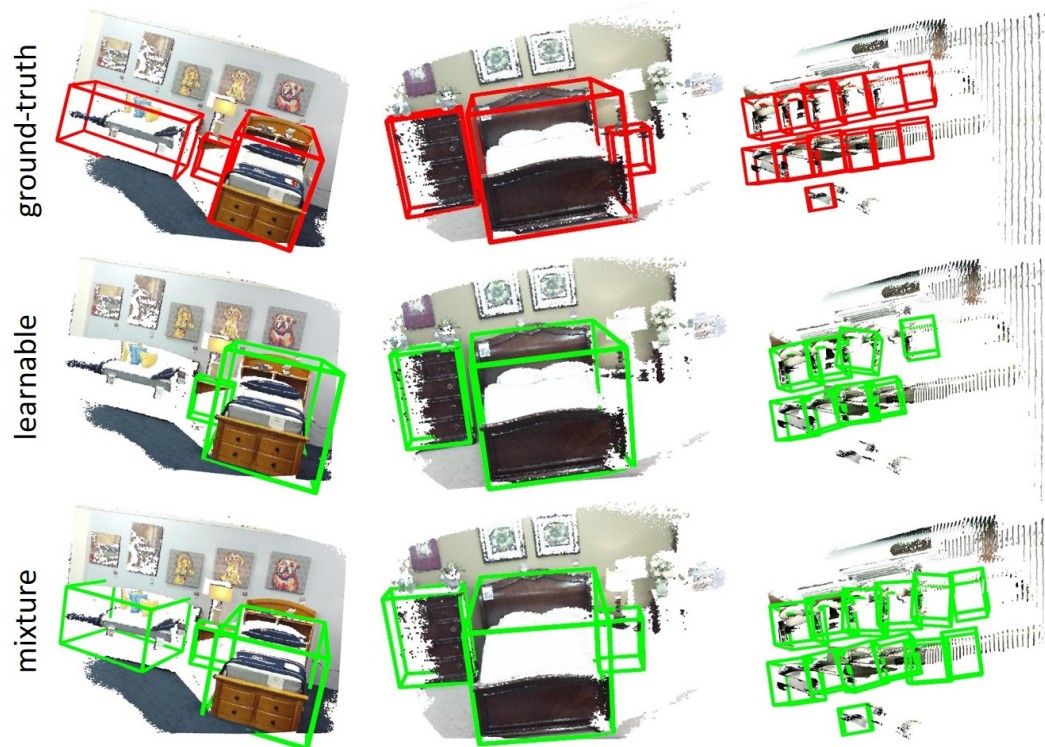

Figure 6: The comparative visualized results of our mixture of query points, compared to the result obtained from learnable query points only. The examples are from the SUN RGB-D dataset.

detector. After we introduce the mixture of query points, global information is better considered with the help of non-learnable query points. As a result, even under the circumstance of insufficient point cloud information, our Uni3DETR can still recognize these objects and detect them out relying on the knowledge about the whole scene. More visualized examples are also provided in Fig. 7.

**Comparison results about the decoupled IoU.** We then compare the visualized results of Uni3DETR with the normal 3D IoU and plot the results in Fig. 8. It can be seen that when supervising the detector with the normal 3D IoU, although the 3D detector has the ability to detect the foreground objects out, the localization precision remains significantly low. For example, for the second case, two chairs are indeed detected, but the overlaps between the detected instances with the corresponding objects are minimal. Furthermore, the low degree of the overlapped area also results in many duplicated boxes, especially for the above one. The same thing also occurs at the third case. For the first case, besides the localization error, only one chair is detected out. This is because 3D IoU is hard to optimize

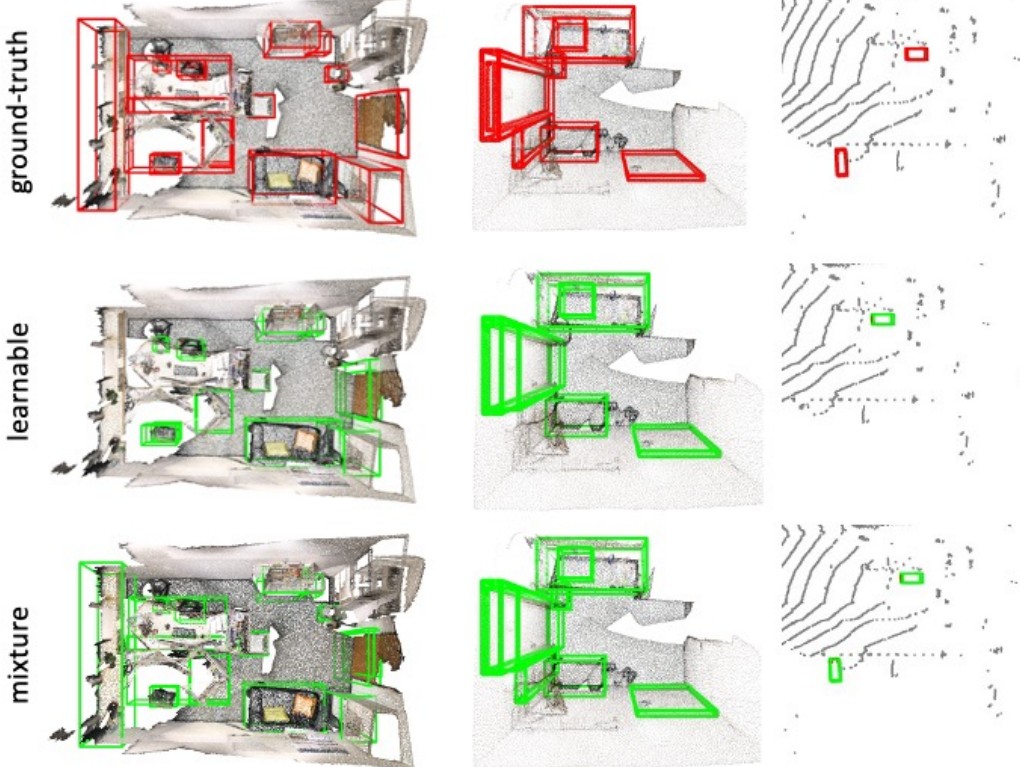

Figure 7: **More comparative visualized results of our mixture of query points, compared to the result obtained from learnable query points only.** The left two examples are from the ScanNet dataset and the right is from the KITTI dataset.

thus fails to alleviate the scale problem of L1 loss. In comparison, our decoupled IoU addresses the coupling problem of 3D IoU, thus contributing to better localization accuracy.

**More visualized results obtained by Uni3DETR.** We further provide more visualized results obtained by our Uni3DETR on different datasets. Uni3DETR obtains satisfying detection results on all five datasets, which further demonstrate its effectiveness and universality.

## D  Per-category Results

Table 13: Per-category $AP_{25}$ for the 10 classes on the SUN RGB-D dataset.

|  | bathtub | bed | bookshelf | chair | desk | dresser | nightstand | sofa | table | toilet | mAP |
|---|---|---|---|---|---|---|---|---|---|---|---|
| H3DNet [70] | 73.8 | 85.6 | 31.0 | 76.7 | 29.6 | 33.4 | 65.5 | 66.5 | 50.8 | 88.2 | 60.1 |
| BRNet [9] | 76.2 | 86.9 | 29.7 | 77.4 | 29.6 | 35.9 | 65.9 | 66.4 | 51.8 | 91.3 | 61.1 |
| GroupFree [30] | 80.0 | 87.8 | 32.5 | 79.4 | 32.6 | 36.0 | 66.7 | 70.0 | 53.8 | 91.1 | 63.0 |
| FCAF3D [43] | 79.0 | 88.3 | **33.0** | 81.1 | 34.0 | 40.1 | 71.9 | 69.7 | 53.0 | 91.3 | 64.2 |
| **Uni3DETR (ours)** | **80.7** | **89.1** | 30.7 | **85.6** | **38.6** | **42.7** | **74.7** | **75.1** | **59.2** | **93.9** | **67.0** |

Table 14: Per-category $AP_{50}$ for the 10 classes on the SUN RGB-D dataset.

|  | bathtub | bed | bookshelf | chair | desk | dresser | nightstand | sofa | table | toilet | mAP |
|---|---|---|---|---|---|---|---|---|---|---|---|
| H3DNet [70] | 47.6 | 52.9 | 8.6 | 60.1 | 8.4 | 20.6 | 45.6 | 50.4 | 27.1 | 69.1 | 39.0 |
| BRNet [9] | 55.5 | 63.8 | 9.3 | 61.6 | 10.0 | 27.3 | 53.2 | 56.7 | 28.6 | 70.9 | 43.7 |
| GroupFree [30] | 64.0 | 67.1 | 12.4 | 62.6 | 14.5 | 21.9 | 49.8 | 58.2 | 29.2 | 72.2 | 45.2 |
| FCAF3D [43] | 66.2 | **69.8** | **11.6** | 68.8 | 14.8 | 30.1 | 59.8 | 58.2 | 35.5 | 74.5 | 48.9 |
| **Uni3DETR (ours)** | **67.4** | 66.2 | 10.7 | **71.7** | **14.8** | **33.5** | **60.3** | **63.0** | **36.7** | **78.6** | **50.3** |

We first list the per-category results for the 10 classes on the SUN RGB-D dataset in Tab. 13 and Tab. 14. For the $AP_{25}$ metric, Uni3DETR achieves the best for 9 classes out of the total 10 classes. The

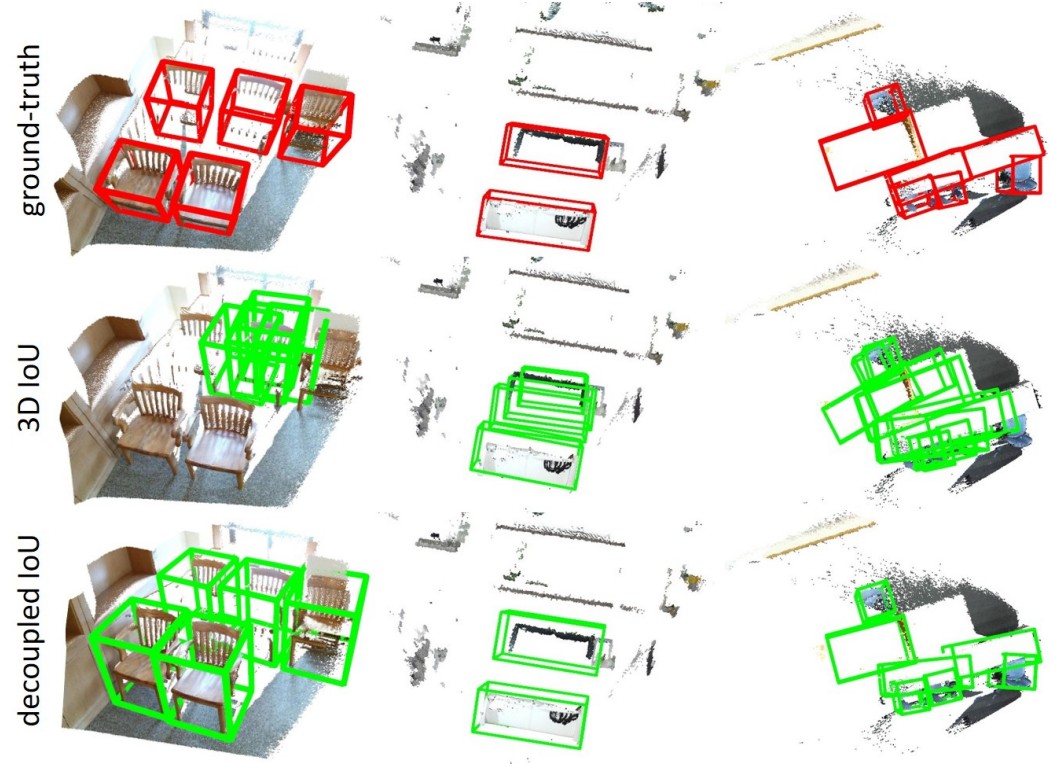

Figure 8: The comparative visualized results of our decoupled IoU, compared to the result obtained from the normal 3D IoU.

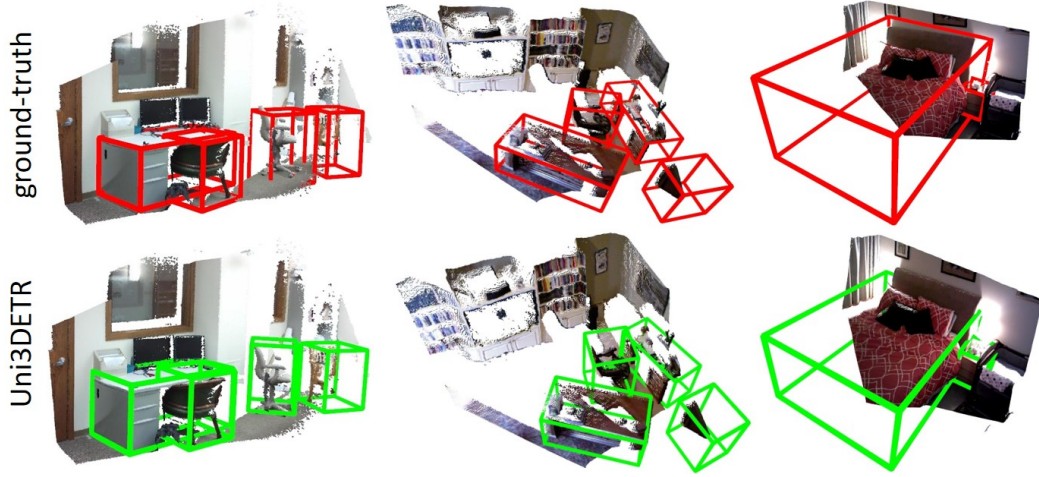

Figure 9: The visualized results of Uni3DETR on the SUN RGB-D dataset.

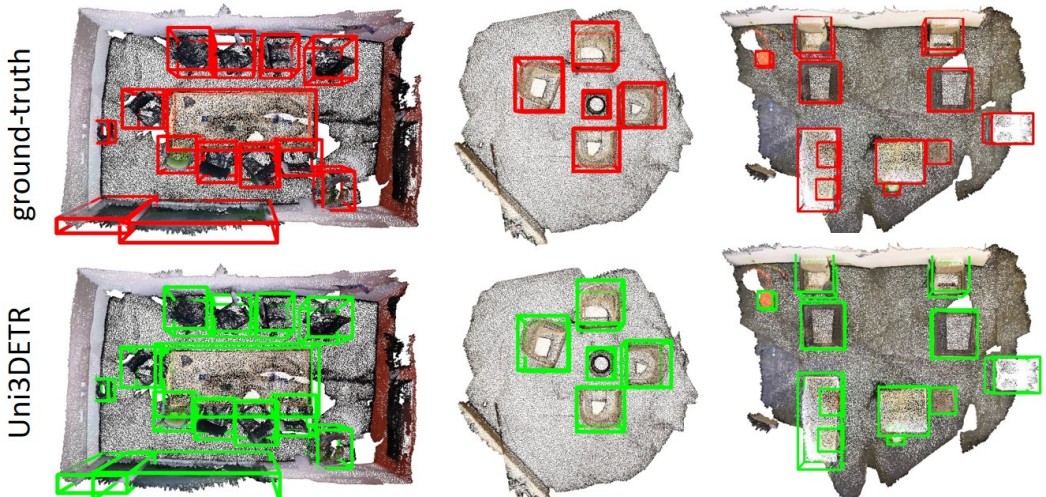

Figure 10: The visualized results of Uni3DETR on the ScanNet dataset.

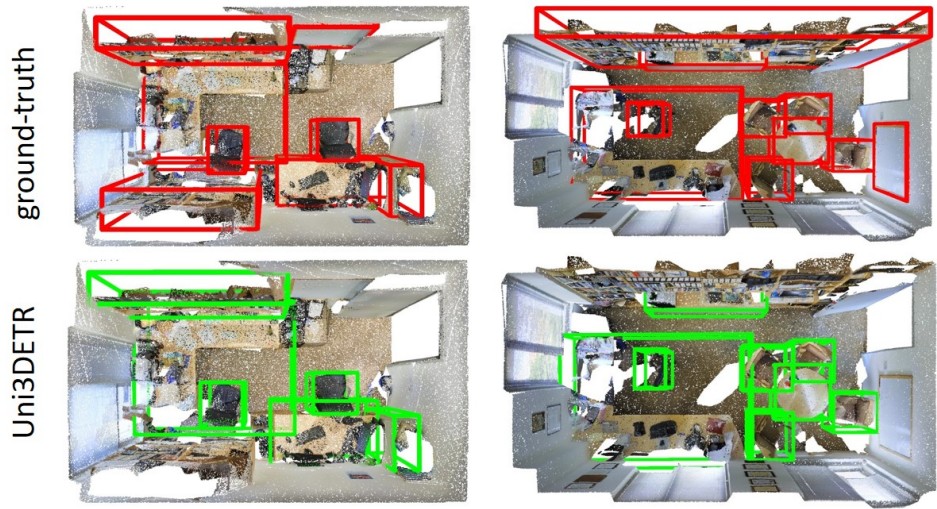

Figure 11: The visualized results of Uni3DETR on the S3DIS dataset.

Table 15: Per-category AP$_{25}$ for the 18 classes on the ScanNet dataset.

| | cab | bed | chair | sofa | tabl | door | wind | bkshf | pic | cntr | desk | curt | fridg | showr | toil | sink | bath | ofurn | mAP |
|---|---|---|---|---|---|---|---|---|---|---|---|---|---|---|---|---|---|---|---|
| VoteNet [38] | 36.3 | 87.9 | 88.7 | 89.6 | 58.8 | 47.3 | 38.1 | 44.6 | 7.8 | 56.1 | 71.7 | 47.2 | 45.4 | 57.1 | 94.9 | 54.7 | 92.1 | 37.2 | 58.7 |
| GSDN [17] | 41.6 | 82.5 | 92.1 | 87.0 | 61.1 | 42.4 | 40.7 | 51.5 | 10.2 | 64.2 | 71.1 | 54.9 | 40.0 | 70.5 | **100** | 75.5 | 93.2 | 53.1 | 62.8 |
| H3DNet [70] | 49.4 | 88.6 | 91.8 | 90.2 | 64.9 | 61.0 | 51.9 | 54.9 | 18.6 | 62.0 | 75.9 | 57.3 | 57.2 | 75.3 | 97.9 | 67.4 | 92.5 | 53.6 | 67.2 |
| GroupFree[30] | 52.1 | **92.9** | 93.6 | 88.0 | 70.7 | 60.7 | 53.7 | 62.4 | 16.1 | 58.5 | 80.9 | **67.9** | 47.0 | 76.3 | 99.6 | 72.0 | **95.3** | 56.4 | 69.1 |
| FCAF3D [43] | 57.2 | 87.0 | **95.0** | **92.3** | 70.3 | 61.1 | **60.2** | **64.5** | 29.9 | 64.3 | 71.5 | 60.1 | 52.4 | **83.9** | 99.9 | **84.7** | 86.6 | **65.4** | 71.5 |
| **Uni3DETR (ours)** | **58.1** | 87.0 | 94.9 | 91.2 | **71.7** | **66.9** | 58.5 | 59.6 | **34.6** | **73.2** | **81.0** | 55.6 | **52.7** | 81.2 | 99.6 | 78.2 | 83.5 | 63.7 | **71.7** |

Table 16: Per-category AP$_{50}$ for the 18 classes on the ScanNet dataset.

| | cab | bed | chair | sofa | tabl | door | wind | bkshf | pic | cntr | desk | curt | fridg | showr | toil | sink | bath | ofurn | mAP |
|---|---|---|---|---|---|---|---|---|---|---|---|---|---|---|---|---|---|---|---|
| VoteNet [38] | 8.1 | 76.1 | 67.2 | 68.8 | 42.4 | 15.3 | 6.4 | 28.0 | 1.3 | 9.5 | 37.5 | 11.6 | 27.8 | 10.0 | 86.5 | 16.8 | 78.9 | 11.7 | 33.5 |
| GSDN [17] | 13.2 | 74.9 | 75.8 | 60.3 | 39.5 | 8.5 | 11.6 | 27.6 | 1.5 | 3.2 | 37.5 | 14.1 | 25.9 | 1.4 | 87.0 | 37.5 | 76.9 | 30.5 | 34.8 |
| H3DNet [70] | 20.5 | 79.7 | 80.1 | 79.6 | 56.2 | 29.0 | 21.3 | 45.5 | 4.2 | 33.5 | 50.6 | 37.3 | 41.4 | 37.0 | 89.1 | 35.1 | 90.2 | 35.4 | 48.1 |
| GroupFree[30] | 26.0 | 81.3 | 82.9 | 70.7 | 62.2 | 41.7 | 26.5 | 55.8 | 7.8 | 34.7 | **67.2** | 43.9 | 44.3 | 44.1 | 92.8 | 37.4 | **89.7** | 40.6 | 52.8 |
| FCAF3D [43] | 35.8 | 81.5 | 89.8 | **85.0** | 62.0 | 44.1 | 30.7 | **58.4** | 17.9 | 31.3 | 53.4 | **44.2** | **46.8** | 64.2 | 91.6 | **52.6** | 84.5 | **57.1** | 57.3 |
| **Uni3DETR (ours)** | **39.5** | **82.5** | **90.4** | 83.1 | **63.8** | **50.5** | **31.9** | 56.4 | **23.5** | **38.6** | 62.8 | 38.4 | 42.2 | 61.6 | **97.8** | 50.9 | 80.2 | 56.3 | **58.3** |

most significant improvement comes from the sofa and table class, 5.4% and 6.2% respectively. For the AP50 metric, UniDETR also achieves the best for 8 classes. For the sofa class, the improvement is up to 4.8%. The effectiveness of UniDETR is thus further demonstrated.

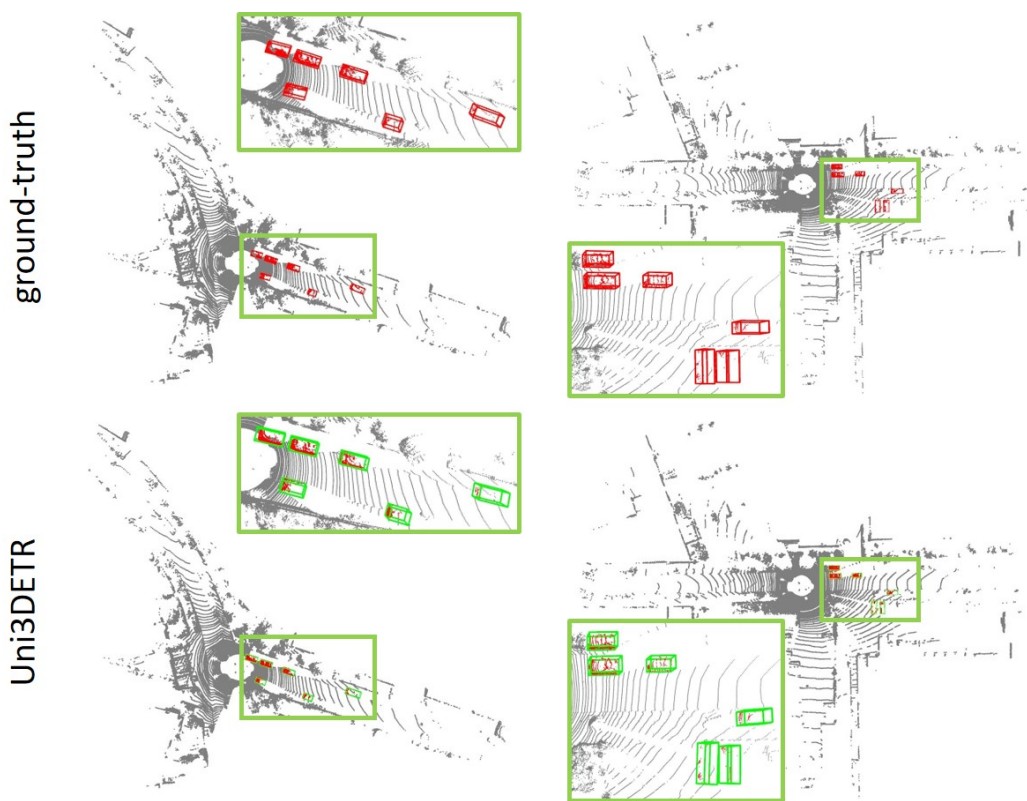

Figure 12: The visualized results of Uni3DETR on the KITTI dataset.

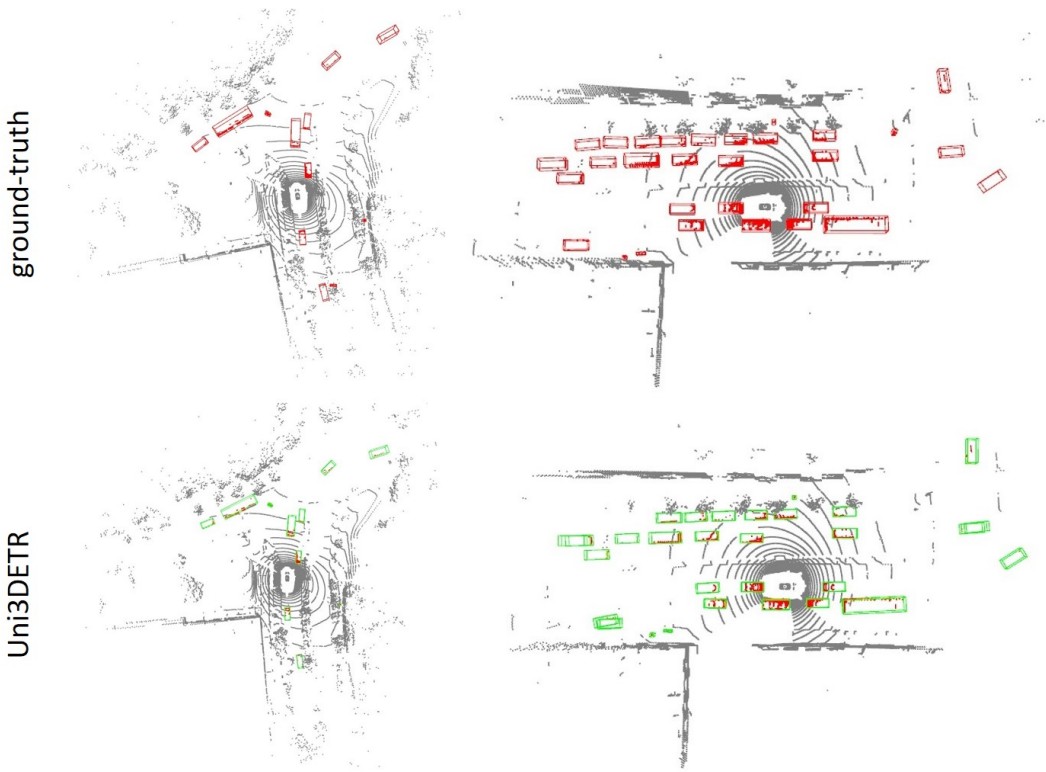

Figure 13: The visualized results of Uni3DETR on the nuScenes dataset.

Table 17: Per-category $AP_{25}$ for the 5 classes on the S3DIS dataset.

| | table | chair | sofa | bkcase | board | mAP |
|---|---|---|---|---|---|---|
| GSDN [17] | 73.7 | 98.1 | 20.8 | 33.4 | 12.9 | 47.8 |
| FCAF3D [43] | 69.7 | 97.4 | **92.4** | 36.7 | 37.3 | 66.7 |
| **Uni3DETR (ours)** | **74.4** | **98.7** | 77.4 | **47.7** | **52.2** | **70.1** |

Table 18: Per-category $AP_{50}$ for the 5 classes on the S3DIS dataset.

| | table | chair | sofa | bkcase | board | mAP |
|---|---|---|---|---|---|---|
| GSDN [17] | 36.6 | 75.3 | 6.1 | 6.5 | 1.2 | 25.1 |
| FCAF3D [43] | 45.4 | **88.3** | **70.1** | **19.5** | 5.6 | 45.9 |
| **Uni3DETR (ours)** | **45.4** | 87.9 | 64.1 | 19.0 | **23.7** | **48.0** |

Table 19: Per-category AP for the 10 classes on the nuScenes dataset.

| | NDS | mAP | Car | Truck | Bus | Trailer | C.V. | Ped | Mot | Byc | T.C. | Bar |
|---|---|---|---|---|---|---|---|---|---|---|---|---|
| UVTR [27] | 67.7 | 60.9 | 85.3 | 53.0 | 69.1 | 41.4 | **24.0** | 82.6 | **70.4** | **52.9** | 67.1 | 63.4 |
| **Uni3DETR (ours)** | **68.5** | **61.7** | **87.0** | **59.0** | **70.8** | **41.7** | 23.9 | **86.1** | 66.4 | 46.0 | **67.8** | **68.0** |

The per-category results for the 18 classes on the ScanNet dataset are listed in Tab. 15 and Tab. 16. We achieve the best for 7 classes for the $AP_{25}$ metric and for 9 classes for the $AP_{50}$ metric. For the S3DIS dataset, the per-category results are listed in Tab. 17 and Tab. 18. For the $AP_{25}$ metric, we achieve the best for 4 classes out of the total 5 ones. For the $AP_{50}$ metric, we are 18.1% higher than FCAF3D on the board class. These per-category results further demonstrate the ability of Uni3DETR on indoor scenes.

We also list the per-category AP on the nuScenes dataset in Tab. 19. Our Uni3DETR achieves the best for 7 out of 10 classes. The ability of Uni3DETR is further demonstrated on outdoor scenes.

