# OpenReview forum: "Uni3DETR: Unified 3D Detection Transformer"
_NeurIPS.cc/2023/Conference — NeurIPS 2023 poster_

### Official Review · Reviewer_Q3d3 · 2023-07-05

**Soundness:** 3 good
**Presentation:** 3 good
**Contribution:** 3 good
**Rating:** 6
**Confidence:** 4

**Summary:**

The authors explore and analyze the existing LiDAR-based 3D object detection framework, and propose to adopt transformer-based method to perform detection from differnt LiDAR inputs. The experimental results on a number of datasets are better compared to the existing methods.

**Strengths:**

1. The task of 3D object detection is very important in the 3D community. Interestingly, the authors propose to adopt transformers to generalize 3D detection across diverse scene inputs.

2. The paper is easy to follow.

3. The authors conduct the experiments on widely-used indoor and outdoor datasets, including  SUN RGB-D, ScanNet, S3DIS,m nuScenes, and KITTI.

**Weaknesses:**

1. Missing important SOTA works on KITTI. The manuscript does report validation results on KITTI, however, the authors forgot to compare with LiDAR-only SOTAs 'PV-RCNN++' (PV-RCNN++: Point-Voxel Feature Set Abstraction With Local Vector Representation for 3D Object Detection). However, it would be much convincing if the authors can compare with this SOTA.

2. Results on KITTI and nuScenes test set. In table 2 & 3, the proposed Uni3DETR is comparable/better with the cited methods. However, it would be much convincing if the authors can report the test set results on KITTI and nuscenes.

3. Computation/memory footprint comparison. The authors didn't make a comparison of their work in terms of memory/speed with the existing 3D detection methods. The time consumption might be large since the memory/time are espeically heavy for the transformer-based methods from my point.


**Questions:**

Please refer to the questions that I describe in the Weakness part. I would also consider the rebuttal and other reviews.

**Limitations:**

NA.

---

> ### Author Rebuttal · Authors · 2023-08-08
>
> **Q1: It would be much convincing if the authors can compare with this SOTA (PV-RCNN++)**
>
> A1: To compare with PV-RCNN++, we follow the same setting to train our model on the training and validation set of KITTI, and evaluate on the KITTI test set. The comparison is listed in the below Tab. 5-1. Our method obtains the 82.26% AP on the car category of the KITTI test set, surpassing PV-RCNN++ by 0.38 points. This validates the effectiveness of our method.
>
> Table 5-1: The comparison of Uni3DETR against PV-RCNN++ on the KITTI test set with 40 recall positions.
> |  |AP-car|
> | :--: | :--: |
> |PV-RCNN|81.43|
> |PV-RCNN++|81.88|
> |ours|82.26|
>
> **Q2: It would be much convincing if the authors can report the test set results on KITTI and nuscenes.**
>
> A2:
>
> We conduct the experiment and evaluate our method on the KITTI test set with 40 recall positions. The comparison is listed in the below Tab. 5-2. For the most important KITTI metric, AP on the moderate level of car, we obtain the 82.26% AP, which is 0.83 points higher than PV-RCNN and 0.49 points higher than CT3D.  The consistent superiority further demonstrates the ability of Uni3DETR on outdoor 3D detection.
>
> Due to time constraints, we have not yet been able to provide results on the NuScenes test set. We will include such experimental comparisons in the final version.
>
> Table 5-2: The performance of Uni3DETR for outdoor 3D object detection on the KITTI test set with 40 recall positions. *: AP on the moderate car is the most important metric for KITTI.
> | |easy|moderate*|hard|
> | :--: | :--: | :--: | :--: |
> |SECOND|88.61|78.62|77.22|
> |PointPillar|82.58 |74.31|68.99|
> |Part-A2|87.81|78.49 |73.51|
> |PV-RCNN|90.25 |81.43 |76.82|
> |CT3D|87.83 |81.77 |77.16|
> |ours|91.14|82.26|77.58|

---

> > ### Comment · Area_Chair_qrjY · 2023-08-20
> >
> > Dear reviewer,
> >
> > Please look over the author response and the other reviews and update your opinion.  Please ask the authors if you have additional questions before the end of the discussion period.

---

> > ### Comment · Reviewer_Q3d3 · 2023-08-21
> > **Authors have addressed most of my concerns.**
> >
> > Thanks for the answers and clarification in the rebuttal, which covered most of my concerns.

---

> > > ### Author Response · Authors · 2023-08-22
> > > **Thanks for your time**
> > >
> > > We sincerely thank your feedback and support. We will polish the paper and add the experimental comparisons following your suggestions. Your constructive comments will help improve the quality of this work.

---

### Official Review · Reviewer_r5Yb · 2023-07-05

**Soundness:** 2 fair
**Presentation:** 3 good
**Contribution:** 2 fair
**Rating:** 5
**Confidence:** 5

**Summary:**

The authors propose a  unified detr-style detector for both the indoor and outdoor 3D object detection. Besides common learnable query, they also adopt unlearnanle query sampled from raw point cloud.

**Strengths:**

The proposed method is simple and easy to follow.

**Weaknesses:**

The novelty is limted, and the experimental results are not fair/convincing.

**Questions:**

1. Uni3DETR follows the design of DETR, which is a ''sparse'' and end-to-end detector. However, Uni3DETR  uses 4 groups of query points for indepedent prediction and needs extra pos-procedure to filer redundant boxes.  The authors claim that the learnable/unlearnable queries capture the local and global information of the scene.  If so many different queries are neccesary to cover the whole scene, why not use a dense detector like pointpillar or centerpoints instead?
2. I am surprised that the results in Tab. 6. I hope the authors could provide more explanation about why multiple groups of queries brings no performance gain?  Why this ablation is not conducted on other datasets (e.g. KITTI)  like Tab.5 and Tab. 7?
3. Using 4 groups of query points (learnable  & unlearnable) as proposed in the paper, or simply use more learnable queries, which one is better?
4. The authors are suggestd to provide results on the KITTI and nuscenes test split. Besides,  KITTI provides a more fair and relibale evaluation setting with 40 recall positions.
5. What is the diffence between Uni3DETR and DAB-Deform in Tab. 4.
6. With differenent backbone, model size, inference speed, ect., it is not fair for the comparison with other sota methods in Tab.2 & Tab.3.


**Limitations:**

The novelty is limited, and more comprehensive analysis is required.

---

> ### Author Rebuttal · Authors · 2023-08-08
>
> **Q1: Why not use a dense detector like pointpillar or centerpoints instead?**
>
> A1:
> * The main reason is that these dense detectors are usually based on the 3D convolution structure and 3D anchors. They perform 3D box generation directly on the extracted features, thus are sensitive to the distinction from the point cloud data. As a result, they cannot perform 3D indoor and outdoor detection with the same structure. In comparison, our 3D detection transformer directly matches 3D predicted boxes and ground-truth boxes, thus tends to be resistant to the difference from data. Therefore, our architecture is more suitable for 3D detection in various environments.
> * We further list the indoor performance on SUN RGB-D and outdoor performance on KITTI of these methods in the following Tab. 4-1. As these dense detectors are designed for outdoor 3D detection, their performance deteriorates seriously for indoor detection. In comparison, our method obtains a satisfying detection performance for both indoor and outdoor detection.
>
> Table 4-1: The performance comparison on the indoor SUN RGB-D and outdoor KITTI dataset with dense detectors PointPillar and CenterPoint. The metrics are AP25 for SUN RGB-D and AP70 for the KITTI car class.
> | |AP-indoor (SUN RGB-D)|AP-outdoor (KITTI)|
> | :--: | :--: | :--: |
> |PointPillar|N/A|77.6|
> |CenterPoint|18.9|74.4|
> |ours|67.0|86.7|
>
> **Q2: I hope the authors could provide more explanation about why multiple groups of queries brings no performance gain? Why this ablation is not conducted on other datasets (e.g. KITTI) like Tab.5 and Tab. 7?**
>
> A2:
> * The reason is that multiple groups of queries simply utilize local information multiple times, and does not provide complementary information. Therefore, it cannot bring further performance improvement. The reason why multiple groups are effective for 2D detection is mainly that it speeds up convergence, which is one of the main problems of transformer-based models for 2D detection. However, in 3D detection, since 3D voxel features contain stronger spatial information, convergence has not been a major problem for transformer-based 3D detectors. Therefore, multiple groups of queries are no longer effective for 3D detection.
> * To validate this, we further conduct such an ablation study on the KITTI dataset and list the comparison in the Tab. 4-2 below. It can be seen that our mixture of query points helps improve AP-car by 0.67%. However, multiple groups of queries fail to improve the 3D detection performance. Such comparison on the outdoor dataset validates that multiple groups of queries cannot boost the performance of 3D detection.
>
> Table 4-2: Comparison with multiple groups of learnable query points on the KITTI dataset.
> |query|AP-car|
> | :--: | :--: |
> |{$P_l$}|85.59|
> |{$P_l$}x2|85.32|
> |{$P_l$}x3|85.18|
> |{$P_l$}x5|85.43|
> |{$P_l$, $P_{nl}$}|85.94|
> |{$P_l$, $P_{nl}$, $P_{nlv}$}|86.26|
>
> **Q3: Using 4 groups of query points (learnable & unlearnable) as proposed in the paper, or simply use more learnable queries, which one is better?**
>
> A3: We conduct the experiment on the SUN RGB-D dataset by simply using 4 times of learnable queries, and list the comparison of learnable query only and our mixture of query points in the following Tab. 4-3. It can be seen that by using more learnable queries, both AP25 and AP50 decrease.  Using more learnable queries cannot provide complementary information, and will even make it more difficult for Hungarian matching. More false positive examples may thus appear, which hurt the performance. In comparison, by using the mixture of query points, we comprehensively utilize the global and local information, thus is better for 3D object detection.
>
> Table 4-3: Comparison with more learnable queries on the SUN RGB-D dataset.
> ||AP25|AP50|
> | :--: | :--: | :--: |
> |learnable only|62.6|46.4|
> |query number x 4|61.7|43.6|
> |learnable + non-learnable|67.0|50.3|
>
> **Q4: The authors are suggested to provide results on the KITTI and nuscenes test split. Besides, KITTI provides a more fair and reliable evaluation setting with 40 recall positions.**
>
> A4:
>
> We conduct the experiment and evaluate our method on the KITTI test set with 40 recall positions. The comparison is listed in the below Tab. 4-4. For the most important KITTI metric, AP on the moderate level of car, we obtain the 82.26% AP, which is 0.83 points higher than PV-RCNN and 0.49 points higher than CT3D.  The consistent superiority further demonstrates the ability of Uni3DETR on outdoor 3D detection.
>
> Due to time constraints, we have not yet been able to provide results on the NuScenes test set. We will include such experimental comparisons in the final version.
>
> Table 4-4: The performance of Uni3DETR for outdoor 3D object detection on the KITTI test set with 40 recall positions. *: AP on the moderate car is the most important metric for KITTI.
> | |easy|moderate*|hard|
> | :--: | :--: | :--: | :--: |
> |SECOND|88.61|78.62|77.22|
> |PointPillar|82.58 |74.31|68.99|
> |Part-A2|87.81|78.49 |73.51|
> |PV-RCNN|90.25 |81.43 |76.82|
> |CT3D|87.83 |81.77 |77.16|
> |ours|91.14|82.26|77.58|
>
> **Q5: What is the difference between Uni3DETR and DAB-Deform in Tab. 4.**
>
> A5: The detection transformer in Uni3DETR takes 3D points as the input of the deformable attention, while DAB-Deform takes 3D anchor boxes (7 dims) as the input. The superiority of taking 3D points as input is mainly twofold. The first is that 3D anchors are negatively affected by the center point missing in 3D point clouds, which makes formulating queries as 3D anchors less effective. In comparison, formulating queries as 3D points better adapts the structure of 3D data, thus achieving better performance. The second is that this accommodates our designed mixture of query points, since FPS points can directly forward into the detection transformer. The mixture of query points can bring further improvement. Ultimately, our Uni3DETR achieves the 66.4% AP (v.s. the 62.0% AP from DAB-Deform).

---

> > ### Comment · Reviewer_r5Yb · 2023-08-18
> > **Response to authors**
> >
> > Thanks, the response addresses my concerns.

---

> > > ### Author Response · Authors · 2023-08-20
> > > **Thanks for your time**
> > >
> > > We sincerely thank your feedback and support. We will polish the paper and rewrite confusing parts following your suggestions. Your constructive comments will help improve the quality of this work.

---

### Official Review · Reviewer_iCdw · 2023-07-06

**Soundness:** 3 good
**Presentation:** 3 good
**Contribution:** 2 fair
**Rating:** 5
**Confidence:** 5

**Summary:**

Uni3DETR is a unified 3D object detector that is capable of handling both indoor and outdoor scenes within the same framework. This is significant as many existing detectors are specialized for either indoor or outdoor environments, but not both.

The method employs a detection transformer with a point-voxel interaction mechanism and a mixture of query points. This allows it to exploit both global and local information effectively.

Uni3DETR introduces a decoupled Intersection over Union (IoU), which serves as an easy-to-optimize training target for localization. This is aimed at improving the accuracy of the detector.

The paper demonstrates that Uni3DETR exhibits excellent performance consistently on both indoor and outdoor 3D detection tasks. Moreover, it shows strong generalization ability under heterogeneous conditions, which is beneficial for real-world applications.

**Strengths:**

Versatility Across Environments: One of the major strengths of Uni3DETR is its ability to handle both indoor and outdoor environments within the same framework. This is a significant advancement as it eliminates the need for different models for different environments, which is common in existing approaches.

Optimization of Localization: The introduction of a decoupled Intersection over Union (IoU) as a training target for localization is a notable strength. This provides an easy-to-optimize objective that can potentially lead to more accurate localization of objects, which is often a challenging aspect of object detection.

Empirical Validation: The paper provides empirical evidence showing that Uni3DETR consistently performs well on both indoor and outdoor 3D detection tasks. This is important for establishing the credibility and effectiveness of the proposed approach.

**Weaknesses:**

Innovative Element. The core pipeline appears to be a Deformable DETR complemented by non-learnable queries (voxel features derived from FPS). It essentially resembles a simple combination of two model architectures.

Efficiency. The model's efficiency is not entirely at the cutting-edge level. For instance, in nuScenes, several competitive 3D detection techniques are not listed in your tables. (https://www.nuscenes.org/object-detection?externalData=all&mapData=all&modalities=Lidar).

Increase in Precision. According to Table 5, the non-learnable query appears to have a marginal impact on outdoor detection. This aligns with our prior understanding that FPS is not well-suited for outdoor detection. The mixed query's design might be superfluous for outdoor scenarios.

Computational Complexity. It is widely recognized that greater computation often results in superior performance. The authors should include a breakdown of the mixed query's complexity, such as inference latency and FLOPS, in comparison to other techniques.

**Questions:**

As evidenced in Table 5, the non-learnable query makes a small contribution to outdoor detection. Can you provide a comparable analysis of the impact of the mixed query points on nuScenes? I'm curious to determine if the performance on nuScenes is primarily driven by your learnable query.

Could you supply a comparison of computational complexity and inference latency between your approach and alternative methods?

**Limitations:**

Integrating two distinct 3D detection algorithms in this method might potentially increase computational complexity. Nonetheless, computation overhead is a key factor for outdoor detection tasks, such as autonomous driving. Given this, the inclusion of a non-learnable query design seems to be superfluous in such outdoor environments.

---

> ### Author Rebuttal · Authors · 2023-08-08
>
> **Q1: It essentially resembles a simple combination of two model architectures.**
>
> A1:
> * Our main contribution is that we propose a unified architecture to address both indoor and outdoor 3D detection within the same framework. Unlike previous methods, which consider indoor and outdoor 3D detection as two problems and address them with a separate set of benchmarks, we are the first to build a unified architecture and eliminate the need for different models in different environments. This is the main novelty of our work. Through Uni3DETR, it is demonstrated that although significant distinction exists for indoor and outdoor point clouds, a unified structure is still possible. Therefore, our framework can be heuristic and become a basic platform or foundation for future works.
> * A simple combination of two model architectures cannot address the problem of for both indoor and outdoor 3D detection. The deformable DETR is built on learned reference points, thus cannot take non-learnable points as the input. We thus modify its structure to directly take 3D points as the input, and perform cross-attention with voxel features. In addition, previous indoor detectors usually utilize FPS for learning point-wise features. Different from them, we utilize FPS points to remedy global information for learning voxel-wise features, which are more resistant to the data distinction. The learnable and non-learnable query points will interact with each other through attention, for a comprehensive understanding of both global and local information. By focusing on information of different levels, the mixture of query becomes a necessary condition for the success of both indoor and outdoor detection.
> * We also propose a decoupled IoU to provide an easy-to-optimize objective for localization. As can be seen from the ablation study, the decoupled IoU improves a lot for both indoor and outdoor detection, which is thus also a necessary element of our framework and a main contribution to the community for unifying indoor and outdoor detection with the same model.
>
> **Q2: In nuScenes, several competitive 3D detection techniques are not listed in your tables.**
>
> A2: Actually many methods in the nuScenes leaderboard apply many techniques, like larger and more complicated backbones or multiple model ensembling. To evaluate different methods, we should put them into the same setting. Under the same setting and comparing in a fair way, our method is actually better. Take the Real-Aug++ method (top 1 in the leaderboard) for example. It obtains the 81.7% AP on the car category of KITTI validation. In comparison, we obtain the 86.6% AP. The effectiveness of our method can thus be validated.
>
> **Q3. The mixed query's design might be superfluous for outdoor scenarios.**
>
> A3:
> * Indeed, introducing non-learnable query points is more effective for indoor 3D detection, as global information is more important for dense small-range indoor scenes. However, what we want is a unified architecture for both indoor and outdoor detection. Without the non-learnable query, the performance of 3D indoor detection will be limited because of the insufficiency of global information. Therefore, although the mixture of query points improves less for outdoor scenarios, it is still necessary.
> * In some certain situations of outdoor scenes, like small, unclear, or occluded objects, local information only will be insufficient, and the non-learnable query will become more effective. To demonstrate this, we conduct the 3D detection experiment on the 3 classes of KITTI (Tab. 3-1). The mixture of query points improves more for the pedestrian and cyclist class, 2.05% for pedestrian and 3.19% for cyclist. Therefore, although the mixed query is less effective for KITTI car, it is still effective for those small and occluded objects.
> * We also provide one visualized example from KITTI in the rebuttal PDF file. As can be seen from the third example of Fig. 1, the available points for the left car are insufficient, which makes it difficult to be detected. In this situation, by leveraging a global understanding of the scene, the non-learnable query helps detect it. Therefore, the non-learnable query is still effective for certain outdoor situations.
>
> Table 3-1: Effect of the mixture of query points on the 3 classes of the KITTI dataset.
> | |AP-car|AP-ped|AP-cyc|
> | :--: | :--: | :--: | :--: |
> |learnable query only|85.24|60.44|69.71|
> |mixture of query points|86.57|62.49|72.90|
>
> **Q4: The authors should include a breakdown of the mixed query's complexity**
>
> A4: The time consumption during inference (latency) retains nearly the same, and the FLOPS also does not increase too much. This demonstrates that the mixture of query points does not bring too much complexity, thus will not bring extra computation overhead.
>
> Table 3-2: The complexity analysis about the mixture of query points.
> | |latency (s)|FLOPS (G)|
> | :--: | :--: | :--: |
> |w/o mix query|0.51|452.23|
> |w/ mix query|0.52 (+ 1.9%)|458.74 (+1.4%)|
>
> **Q5: Can you provide a comparable analysis of the impact of the mixed query points on nuScenes?**
>
> A5: As can be seen, the mixture of query points brings a 1.9% mAP improvement. The improvement is a little more than KITTI, because nuScenes consists of more small objects. We further select four classes and list their per-category AP. For the car category, the AP improvement is 1%. For other categories with usually small or occluded objects, the improvement is more significant: the mixture of query points boost the pedestrian class by 2.1%, the motorcycle class by 3.6%, and 4% for the traffic cone class. In a word, the non-learnable query is equally effective for these small or occluded objects in outdoor scenes.
>
> Table 3-3: Effect of the mixture of query points on the nuScenes dataset.
> | |mAP|car|pedestrian|motorcycle|traffic cone|
> | :--: | :--: | :--: | :--: | :--: | :--: |
> |learnable query only|59.8|85.6|84.3|64.0|66.0|
> |mixture of query points|61.7|86.6|86.4|67.6|70.0|

---

> > ### Comment · Area_Chair_qrjY · 2023-08-20
> >
> > Dear reviewer,
> >
> > Please look over the author response and the other reviews and update your opinion.  Please ask the authors if you have additional questions before the end of the discussion period.

---

> > ### Comment · Reviewer_iCdw · 2023-08-20
> >
> > Thanks for the author's comprehensive response! These address my concerns. This work may have practical value in autonomous agents that should both handle the indoor and outdoor scenes.

---

> > > ### Author Response · Authors · 2023-08-20
> > > **Thanks for your time**
> > >
> > > We sincerely thank your feedback and support. We will polish the paper and add necessary discussions following your suggestions. Your constructive comments will help improve the quality of this work.

---

### Official Review · Reviewer_bu7C · 2023-07-08

**Soundness:** 2 fair
**Presentation:** 3 good
**Contribution:** 2 fair
**Rating:** 6
**Confidence:** 4

**Summary:**

The paper proposes Uni3DETR, a unified architecture suitable for indoor and outdoor scenes. Uni3DETR employs the DETR with point-voxel interaction for object prediction. It uses a mixture of query points to exploit global and local information. Finally, Uni3DETR uses a decoupled IoU loss by disentangling the depth and other dimensions to ease out training. Training a per-dataset model and testing on the same dataset improves over the baselines.

**Strengths:**

+ The problem of using a unified architecture for indoor and outdoor point clouds is super relevant.
+ The idea of employing DETR, a mixture of query points, and decoupling 3D IoU is nice.
+ The per-dataset results are good.

**Weaknesses:**

- The first claim that the architecture is uniform is only partially correct. L228 says that the authors chose a 0.02m grid size for indoor datasets, while they chose a different voxel size for the KITTI and the nuscenes datasets. The authors should pick a single voxel size and then run it on all the datasets. (It is OK to change the range for nuScenes since it is a multi-camera dataset, but the authors should choose a single voxel size)

- The claim of "strong universal ability of our model (L336)" is an overstatement, in my opinion, since the paper only carries out Per-dataset model training and testing. A related paper that trains a single RGB 3D detector is Omni3D + Cube R-CNN [A]. I would urge the authors to follow a similar protocol where the train and test sets are
   - KITTI + nuScenes front and then test on indoor indoor/outdoor
   - Train on indoor scenes and then test on outdoor indoor/outdoor
   - Finally, train on all indoor/outdoor scenes, then test on indoor/outdoor scenes.

By training Uni3DETR on KITTI + nuScenes front and testing outdoors, one can compare against the Cube R-CNN model of [A].

- How well does the method generalize in zero-shot settings? E.g., How does the Uni3DETR model trained on outdoor scenes work on an unknown lidar scan from an unknown say, Waymo dataset.

- A sad part of 3D detection papers is to use a bigger backbone and show improvements on nuScenes. Please quantitatively compare the flops, model size, training time, and inference time for Uni3DETR against the baselines in Table 3.

- The authors say the learnable query points mostly contain local information and fit the outdoor detection well. In contrast, non-learnable query points emphasize global information and are more effective for dense indoor scenes. Another way to enforce local and global information is to use a dual-path architecture where one branch is attention-based for global information while the other is convolution-based for local information. One can later fuse their outputs by concatenation or predicting a coefficient to add them. Why did the authors not consider this design choice?

- An alternative to using the decoupled IoU loss is the disentangled loss [B], where one constructs a 3D box from GT parameters except the one in consideration. That way, one can avoid explicitly weighing the losses for respective parameters (e.g., the authors use weighing xy and z terms by 0.5). The authors should, therefore, quantitatively compare the decoupled IoU loss against the disentangled loss.

- The details of the 3D feature extractor (Section 2.1) need to be included.

- The authors use the mean average precision (mAP) under IoU thresholds of 0.25 and 0.5 for evaluating datasets  (L216). Using one or two thresholds is quite sensitive. A better alternative is to use exact IoU3D over a series of thresholds of 0.05:0.5:0.05 and then average them as in [A].

Minor:
- Do the authors plan to release the code?

- Typo: L203: final

References:

A. Omni3D: A Large Benchmark and Model for 3D Object Detection in the Wild, Brazil, Kumar, Ravi, et al., CVPR 2023.

B. MonoDIS: Disentangling monocular 3D object detection: From single to multi-class recognition, Simonelli, Bulo, Porzi, et al., PAMI 2020.

**Questions:**

Please see the weakness.

**Limitations:**

The authors do not list out the limitations.

---

> ### Author Rebuttal · Authors · 2023-08-08
>
> **Q1. The claim that the architecture is uniform is partially correct.**
>
> A1:
> * The word “unified” in our paper specifically refers to the architecture aspect. The voxel size is a data-related parameter, not architecture-related. Since point clouds are collected with different sensors, their ranges and distributions vary significantly (about 3m for indoor but more than 70m for outdoor datasets). Here we follow the settings of previous works and utilize the same grid size. Then 3D voxels can be processed with a unified architecture. We are the first to build a unified 3D detector for both indoor and outdoor point clouds with the same architecture.
> * We also conduct the experiment with the same KITTI voxel size. The resolution of indoor voxels deteriorates, which hurts the detection. However, compared with other outdoor detectors, our superiority is still obvious. Therefore, even if standardizing such a data-related parameter, our model can still make a higher AP.
>
> Table 2-1: 3D detection with the same voxel size of (0.05m, 0.05m, 0.1m).
> | |AP-indoor (SUN RGB-D)|AP-outdoor (KITTI car)|
> | :--: | :--: | :--: |
> |3DSSD|9.5|78.6|
> |CenterPoint|18.9|74.4|
> |UVTR |35.9|72.0|
> |ours (a single voxel size)|47.3|86.7|
> |ours (different voxel sizes)|67.0|86.7|
>
> **Q2: The claim of "strong universal ability of our model" is an overstatement**
>
> A2:
> * The “universality ability” here refers specifically to our construction of a unified architecture for both indoor and outdoor detection. In the current point cloud based 3D detection research, indoor and outdoor detectors are still accomplished with totally distinct structures, and a unified architecture for per-dataset experiments is still absent. From this perspective, the universality ability of our detector is at least better than existing 3D detectors. We will revise it to “Extensive experiments demonstrate that our model can address both indoor and outdoor 3D detection with a unified structure” in the final version.
> * Cube RCNN only takes RGB images for 3D detection. Compared with 2D images, the dataset-interference issue is extremely serious for point clouds. More cross-dataset discrepancies, including sensor type differences (the RGB-D sensor v.s. LiDAR), scene changes (small-range and dense indoor v.s. large-range and sparse outdoor), object distribution (more sparse and small-size in outdoor scenes), make the joint combination of multiple point cloud datasets extremely challenging. As a result, there is still no work for joint training on multiple point cloud datasets. Actually, if we want one model like Cube RCNN but taking point clouds as input, a unified structure should inevitably serve as its prerequisite and foundation. Here we propose a unified architecture for different point clouds, as the foundation towards this. How to bridge the cross-dataset difference for point clouds is not our focus.
> * We also follow a similar protocol to compare with Cube RCNN - training on KITTI and nuScenes then evaluating on KITTI. We observe that even if we do not work on the cross-dataset difference, the performance is still better.
>
> Table 2-2: Comparison by training on KITTI and nuScenes front, then evaluating on KITTI.
> |  |AP-car |
> | :--: | :--: |
> |Cube RCNN|15.0|
> |ours|65.3|
>
> **Q3: How well does the method generalize in zero-shot settings**
>
> A3: We adopt two protocols: training on SUN RGB-D then evaluating on ScanNet; training on KITTI then evaluating on waymo. Its zero-shot AP on ScanNet surpasses GroupFree by 6.2%. Its outdoor zero-shot AP is also 5.2% better than existing outdoor detectors. This well demonstrates its generalization ability.
>
> Table 2-3: 3D detection on ScanNet of SUN RGB-D trained detectors.
> | |AP25|AP50|
> | :--: | :--: | :--: |
> |VoteNet|52.5|31.7|
> |GroupFree3D|51.2|23.3|
> |ours|57.4|38.6|
>
> Table 2-4: 3D detection on waymo of KITTI trained detectors.
> | |AP-car|
> | :--: | :--: |
> |SECOND|5.8|
> |PointPillar|12.1|
> |Part-A2|14.9|
> |ours|20.1|
>
> **Q4: Why did the authors not consider the dual path?**
>
> A4:
> * Convolution-based structures perform 3D box generation directly on the extracted features, thus are sensitive to data distinctions. Such structures thus cannot extract local information well for both indoor and outdoor scenes. We conduct the experiment, where the convolution branch is from Part-A2. Our model outperforms it by more than 10% for both indoor and outdoor datasets. This explains why we do not adopt the dual-path structure.
> * A dual-path architecture will also inevitably bring in more computation budget.
>
> Table 2-5: Comparison with the dual-path structure.
> | |AP-indoor (SUN RGB-D) |AP-outdoor (KITTI car)|
> | :--: | :--: | :--: |
> |dual-path |54.2 |75.6|
> |ours |67.0|86.7|
>
> **Q5: The authors should compare against the disentangled loss**
>
> A5: Our decoupled IoU is 12.5% higher than disentangled loss. The reason is that the original disentangled loss takes RGB images as input, and point clouds only will hurt its performance.
>
> Table 2-6: Comparison on KITTI between our decoupled IoU and the disentangled loss.
> | |AP-car|
> | :--: | :--: |
> |disentangled loss |74.24 |
> |decoupled IoU loss |86.74|
>
> **Q6: The 3D feature extractor details need to be included.**
>
> A6: We add the visualization of our 3D feature extractor in Fig. 2 of our rebuttal PDF file.
>
> **Q7: A better alternative is to use IoU3D over a series of thresholds of 0.05:0.5:0.05.**
>
> A7:
>
> * Here for indoor 3D detection, most previous detectors adopt the AP25 and AP50 metrics. For a fair comparison, we adopt the same metrics.
> * We adopt the same metric as Omni3D for evaluation. We observe that the improvement is consistent. This demonstrates that our model achieves better performance for both recognition and localization.
>
> Table 2-7: Comparison on SUN RGB-D with the Omni3D metric.
> | |AP[0.05:0.5:0.05] |
> | :--: | :--: |
> |VoteNet | 53.4|
> |GroupFree3D |56.8|
> |FCAF3D |60.6|
> |ours|64.3|
>
> **Q8: code release and typos**
>
> A8: We will correct typos in the final version and release the code.

---

> > ### Comment · Reviewer_bu7C · 2023-08-18
> > **Response to authors**
> >
> > Thank you, authors, for putting an excellent rebuttal. The benefits of your architecture are not fully visible if the experiments are not cross datasets/cross domains.
> >
> > - I understand that scaling to datasets with different voxel sizes and domains is an issue with lidar-based 3D detection. Why do not we keep a single voxel size and do a quantitative cross-dataset evaluation with the Omni3D metric over all combinations of dataset **within** the indoor and outdoor domains for Uni3DETR and your baselines? (such as KITTI --> NuScenes, nuScenes --> KITTI, ScanNet --> SUN, SUN --> ScanNet) In other words, please compare with Cube R-CNN table 5. The comparison will let us know how good the within-domain generalization is keeping the voxel size unchanged.
> >
> > - If possible, please also quantitatively report the following results with a single voxel size.
> > Train on indoor and test outdoor for Uni3DETR, your lidar baselines, and Cube R-CNN with the Omni3D metric.
> > Train on outdoor and test indoor for Uni3DETR,  your lidar baselines, and Cube R-CNN with the Omni3D metric.
> >
> > - Your Omni3D numbers in Table 2-2 of the rebuttal differ from Table 5 of the Omni3D paper. They report AP on KITTI as 42.5, while you report as 15.0. Why is it so?
> >
> > - It would also be good to quantitatively compare the flops, model size, training time, and inference time for Uni3DETR against the baselines in Table 3.
> >
> > Please note that it is completely OK if the Uni3DETR numbers on some of the experiments are bad. We are not here to compete on benchmarks and penalize for the bad results. All we want is to clearly understand the limitations of the methods and advance science :)

---

> > > ### Author Response · Authors · 2023-08-18
> > > **Thanks for your response**
> > >
> > > We appreciate your reply. We acknowledge that the cross-dataset ability of 3D detectors is an important problem. As a unified architecture is still absent now, this problem is more serious for point cloud based 3D detection. In this paper, we address the problem of the absence of a unified architecture. We are the first to propose such a unified architecture, which can serve as the foundation and prerequisite for cross-dataset generalization. We follow your suggestions to conduct the experiments below. The experimental results (about 10% ~ 30% improvement over Cube RCNN) demonstrate that our unified architecture has the potential for future directions, such as cross-dataset 3D detection. Through the below experiments, the benefits of our architecture can be better illustrated.
> > >
> > > **Q1: a quantitative cross-dataset evaluation**
> > >
> > > We follow your suggestions to conduct the cross-dataset evaluation with the Omni3D metric. For the outdoor setting, we conduct the experiment of KITTI→nuScenes, nuScenes→KITTI, KITTI+nuScenes (OMNI3D_OUT) → KITTI and nuScenes. For the indoor setting, since Cube RCNN does not train on the ScanNet dataset (the ScanNet dataset provides multi-perspective images for one scene, while Cube RCNN cannot fit such images), we conduct the experiment of ScanNet→SUN RGB-D, and compare it with Cube RCNN trained on OMNI3D_IN.
> > >
> > > From the results, it is worth noticing that Uni3DETR has a good cross-dataset generalization ability. The performance is better than Cube RCNN for both indoor and outdoor evaluation. For the indoor SUN RGB-D dataset, the cross-dataset AP (ScanNet→SUN RGB-D) is 16.2% higher than Cube RCNN trained on the SUN RGB-D dataset. For outdoor scenes, our method also surpasses Cube RCNN, more than 30% higher for nuScenes→KITTI and 15% higher for KITTI→nuScenes. The reason is that our Uni3DETR takes point clouds as input for 3D detection, while Cube RCNN takes RGB images for detection. By introducing 3D space information from point clouds, the superiority of a unified architecture for point clouds over Cube RCNN can be demonstrated.
> > >
> > > We further emphasize that cross-dataset evaluation is a more difficult problem for point cloud based 3D object detection, as the dataset-interference issue is more serious for point clouds. Cube RCNN only takes RGB images as input, which helps it avoid the dataset-interference issue of point clouds. This makes it eligible to conduct cross-dataset evaluation experiments. In this work, we build a unified structure for point clouds, which can serve as the prerequisite and foundation of point cloud based cross-dataset experiments. The experimental results below can demonstrate its potential for cross-dataset generalization. We believe our Uni3DETR can become the basic platform and facilitate related research.
> > >
> > > Table 2-8: Cross-dataset performance on the indoor SUN RGB-D dataset compared with Cube RCNN
> > > |Method|Trained on|AP3D-SUN|
> > > | :--: | :--: | :--: |
> > > |Cube RCNN|SUN RGB-D|34.7|
> > > |Cube RCNN|OMNI3D_IN (containing SUN RGB-D)|35.4|
> > > |Uni3DETR|SUN RGB-D|64.3|
> > > |Uni3DETR|ScanNet|50.9|
> > >
> > > Table 2-9: Cross-dataset performance on the outdoor KITTI and nuScenes dataset compared with Cube RCNN.
> > > |Method|Trained on|AP3D-KIT|AP3D-NU|
> > > | :--: | :--: | :--: | :--: |
> > > |Cube RCNN|KITTI|37.1|12.7|
> > > |Cube RCNN|nuScenes|20.2|38.6|
> > > |Cube RCNN|OMNI3D_OUT|42.4|39.0|
> > > |Uni3DETR|KITTI|83.8|19.4|
> > > |Uni3DETR|nuScenes|54.2|57.3|
> > > |Uni3DETR|OMNI3D_OUT|72.3|52.1|
> > >
> > >
> > >
> > > **Q2: Train on indoor and test outdoor, train on outdoor and test indoor**
> > >
> > > Since there are no overlapped categories between indoor and outdoor scenes, both our Uni3DETR and Cube RCNN cannot perform cross-dataset experiments from indoor to outdoor or from outdoor to indoor. Up to now, no works can achieve such a goal, no matter point cloud based or RGB image based models. However, it can be expected that if such datasets can be created, with overlapped categories between indoor and outdoor scenes, our Uni3DETR will continue to demonstrate the corresponding potential.
> > >
> > > **Q3: Your Omni3D numbers in Table 2-2 of the rebuttal differ from Table 5 of the Omni3D paper**
> > >
> > > The reason is that the Table 5 of Omni3D reports AP on KITTI based on the Omni3D metric, while we adopt the AP70 metric on the car category, the official KITTI evaluation metric. The 15.0% AP70 of Cube RCNN comes from the Table 3 of Omni3D. If we utilize the same Omni3D metric, our performance on the KITTI dataset can be 88.7%, as can be seen in the table below. This further demonstrates the effectiveness of our method.
> > >
> > > Table 2-10: Comparison with Cube RCNN by training our Uni3DETR on the joint of KITTI and nuScenes front, then evaluating on the KITTI dataset.
> > > | |AP-car|AP3D (Omni3D metric)|
> > > | :--: | :--: | :--: |
> > > |Cube RCNN (omni3D)|15.0|42.4|
> > > |ours|65.3|88.7|
> > >
> > >
> > > **Q4: It would also be good to quantitatively compare the flops, model size, training time, and inference time for Uni3DETR against the baselines in Table 3.**
> > >
> > > Please refer to the above author's rebuttal to all reviewers.

---

> > > > ### Author Response · Authors · 2023-08-20
> > > >
> > > > Dear Reviewer bu7C,
> > > >
> > > > We are sincerely grateful to you for the precious time you have devoted to reviewing our paper.
> > > >
> > > > We would like to know whether our response has addressed your concerns and if you have the time to provide further feedback on our rebuttal. We are more than willing to engage in further discussion.
> > > >
> > > > Best regards,
> > > >
> > > > Authors of paper 2701

---

> > > > > ### Comment · Reviewer_bu7C · 2023-08-22
> > > > > **Thankyou authors.**
> > > > >
> > > > > I thank the authors for showing cross-dataset experiments. The cross-dataset experiments are nice and make Uni3DETR unique. I, therefore, raise my rating to weak accept.
> > > > >
> > > > > My acceptance is conditional, provided the authors do the following:
> > > > > - Document all pre-processing steps, from downloading datasets to preparing the datasets and environment.
> > > > > - Release all codes, including training, inference, and cross-dataset inference with dataset-specific and Omni3D metrics. The authors should provide a **single bash script to reproduce numbers on all combinations of train datasets, test datasets, models, and metrics**. These steps are highly crucial for the community to build upon this work.
> > > > > - Release All pre-trained models trained individually on one dataset as well as on a combination of datasets with different voxel sizes and unified voxel sizes.
> > > > > - Quantitatively report the performance of models with the same voxel size **and** different voxel sizes in the tables.
> > > > > - Report the cross-dataset experiments in the main paper.

---

> > > > > > ### Author Response · Authors · 2023-08-22
> > > > > > **Thanks for your time**
> > > > > >
> > > > > > We sincerely thank your feedback and support. We will relase the codes and related documents so that our Uni3DETR can become a basic platform for future works. We will also polish the paper following your suggestions. Your constructive comments will help improve the quality of this work.

---

### Official Review · Reviewer_TGtP · 2023-07-10

**Soundness:** 3 good
**Presentation:** 3 good
**Contribution:** 3 good
**Rating:** 6
**Confidence:** 5

**Summary:**

This paper proposes Uni3DETR, a unified 3D detection transformer that addresses indoor and outdoor 3D detection within the same framework. The paper provides a specific analysis of the inconsistency in the structure of current indoor and outdoor scene detection models. Due to the differences in data distribution between indoor and outdoor environments, indoor datasets have a smaller range of point clouds, with objects being closer together and denser, occupying a larger portion of the point cloud. On the other hand, outdoor scenes have smaller and sparser objects, with background point clouds dominating the overall point cloud. Utilizing the mixture of query points which consist of  the learnable and non-learnable query, the detector is able to exploit the global information for dense small-range indoor scenes and local information for large-range sparse outdoor scenes. Besides, decoupled IoU is proposed for easier and faster training target localization optimization  by disentangling the xy and z space.

**Strengths:**

1. The paper provides a detailed analysis of the inconsistent models of indoor and outdoor 3D detectors, with the difference in data distribution being the most important reason. The author explains how to design a unified 3D detector and solves the problem of inconsistent architectures between indoor and outdoor scenes.
2. The overall logic of the paper is clear, and it is well written.
3. Uni3DETR demonstrates the strong generalization ability under heterogeneous conditions while maintaining caomparable performance.

**Weaknesses:**

1. The experiments of model architectures are a bit insufficient. In line 99, the paper mentions, "Then, we convert the extracted sparse features into dense ones and apply 3D dense convolutions for further feature processing." However, no experiments show the performance gain brought by 3D dense convolution to support the claim. Furthermore, will the model architectures vary from datasets? For example, are the numbers of layers different for models trained on KITTI and S3DIS?
2. The author claims that learnable queries primarily capture the local information of the scene, and non-learnable queries usually concentrate more on global information. Although experiments show that combining different queries could bring better performance, there is no evidence in the paper (e.g., visualization or other analysis) to support the claim.

**Questions:**

1. Decoupled IoU is only used in Uni3DETR, and it can be seen from the ablation study that this loss has a significant performance improvement for the detector. However, Decoupled IoU loss was not used in other mothods when compared to other methods. Can Decoupled IoU improve other detectors, such as GroupFree3D, and also bring significant performance improvements?
2. The learnable query mentioned in the paper lacks its initialization process, such as how to obtain $P_q$ and $C_q$ mentioned in line 107.
3. How about the generalization ability of feature extraction based on this structure for indoor scenes? For example, how it performs zero-shot testing by training on SUN RGBD and directly testing the trained model on ScanNet?
5. The author uses a mixture of query points to obtain detection results through different categories and a sufficient number of queries. Will this result in noticeable time consumption during inference?
6. It would be better to illustrate more visualization results to support the claim that the learnable and non-learnable queries have different capabilities.
7. According to the paper: ConQueR: Query Contrast Voxel-DETR for 3D Object Detection: DETRs usually adopt a larger number of queries than GTs (e.g., 300 queries v.s. ∼40 objects in Waymo) in a scene, which inevitably incur many false positives during inference. Could this problem occur in Uni3DETR since many more queries are used in inference?

**Limitations:**

It seems that the paper now provides a unified paradigm for solving 3D object detection tasks of indoor and out door scenes and do not use one model for testing on all the datasets. Furthermore, the model is separately trained with different hyper-parameters on different datasets.

---

> ### Author Rebuttal · Authors · 2023-08-08
>
> **Q1: The experiments of model architectures are a bit insufficient.**
>
> A1:
> * We conduct the experiments on the SUN RGB-D dataset in the below Tab. 1-1. Dense 3D convolutions contribute to the 6.3% AP25 improvement and 10.1% AP50 improvement. Its effectiveness to the performance can be demonstrated. As sparse convolutions are only conducted to the point positions in the 3D space, it will lead to the feature missing problem of center points. Dense convolutions help alleviate this problem, thus improving the detection accuracy.
> * The architecture of our model is totally the same for all different datasets, including indoor datasets SUN RGB-D, ScanNet, S3DIS and outdoor datasets KITTI, nuScenes. We are the first to build a unified 3D detector, which can perform both indoor and outdoor 3D object detection with the same architecture.
>
> Table 1-1: The performance on the SUN RGB-D dataset with or without 3D dense convolution layers.
> |      |  AP25  | AP50 |
> | :--: | :--: | :--: |
> |w/o dense conv | 60.7 | 40.2|
> |w/ dense conv | 67.0 | 50.3 |
>
> **Q2: There is no evidence in the paper (e.g., visualization or other analysis) to support the claim (the mixture of query points).**
>
> A2:
> * We provide visualized results about the mixture of query points in the section 2 of our supplementary material. With only learnable query points, the detector mainly leverages local information to detect objects in the scene. However, some objects, especially partly occluded and with insufficient points, are easily ignored and missed. This is because without enough points, the local information will be insufficient for these objects to be detected. By providing global information, learning with non-learnable query points can better consider the background. As a result, the detector can obtain an overall understanding of the whole scene, and such global understanding can help the model find and detect these objects.  Therefore, with the mixture of query points, more comprehensive detection results can be obtained.
> * We also provide three more visualized results in Fig. 1 of the rebuttal PDF file to better illustrate the effectiveness of our mixture of query points. We will put such a visualized analysis in the final version of our paper.
>
> **Q3: Can Decoupled IoU improve other detectors and also bring significant performance improvements?**
>
> A3: We involve our decoupled IoU into VoteNet and GroupFree3D, and conduct the experiments on the ScanNet dataset. As we can see, Decoupled IoU still helps improve the detection AP: 7.2% for VoteNet and 2% for GroupFree3D, which demonstrates the easy-to-optimize property of decoupled IoU also helps positional information learning in previous detectors. However, the improvement is less than that in our Uni3DETR. This is because our detector is based on the transformer structure, which is seriously affected by the issue of L1 loss for different scales. Therefore, decoupled IoU is more urgently demanded in our transformer-based detector.
>
> Table 1-2: The performance on the ScanNet dataset. We apply our decoupled IoU loss on the VoteNet and GroupFree3D.
> |      |  Decoupled IoU | AP25  | AP50 |
> | :--: | :--: | :--: | :--: |
> |VoteNet |  | 58.6 | 33.5 |
> |   |  √ | 65.8 | 46.2 |
> |GroupFree3D | | 69.1 |52.8|
> |  | √ |71.1 |53.8|
>
> **Q4. The learnable query mentioned in the paper lacks its initialization process, such as how to obtain $P_q$ and $C_q$ mentioned in line 107.**
>
> A4:
> We initialize both the content query $C_q$ and learnable query points $P_q$ with the standard normal distribution, i.e. randomly initializing them using a commonly utilized standard Gaussian distribution with zero mean and unit standard deviation. As these queries will be learned during training, the initialization approach does not affect the ultimate performance too much.
>
> **Q5. How about the generalization ability for indoor scenes? For example, how it performs zero-shot testing by training on SUN RGBD and directly testing the trained model on ScanNet?**
>
> A5:
> We perform inference on the ScanNet dataset with SUN RGB-D trained model, and compare it with two previous methods VoteNet and GroupFree3D. The results show that Uni3DETR can also obtain a satisfying result for this zero-shot testing experiment, obtaining a 57.4% AP25 and 38.6% AP50. Its zero-shot AP on ScanNet surpasses VoteNet by 4.9% and GroupFree by 6.2%.
>
> Table 1-3: The performance on the ScanNet dataset of a SUN RGB-D trained Uni3DETR.
> |      |  AP25  | AP50 |
> | :--: | :--: | :--: |
> | VoteNet | 52.5 | 31.7 |
> |GroupFree3D|51.2|23.3|
> |ours|57.4|38.6|
>
> **Q6. The author uses a mixture of query points. Will this result in noticeable time consumption during inference?**
>
> A6: As the 3D detection transformer is based on the 3D voxel feature after downsampling, the computational budget of the whole detection transformer is actually not large. As a result, the mixture of query points does not affect the inference time consumption too much. We conduct the complexity analysis on the mixture of query points in the below Tab. 1-4. Both the time consumption during inference (latency) and FLOPS retain nearly the same with the mixture of query points. This demonstrates that our mixture of query points does not introduce too much extra computation.
>
> Table 1-4: The complexity analysis about the latency and FLOPS about the mixture of query points.
> |  | latency (s) | FLOPS (G)|
> | :--: | :--: | :--: |
> |w/o mix query | 0.51 |452.23|
> |w/ mix query |0.52 (+ 1.9%) |458.74 (+1.4%)|
>
> **Q7: Could the large number query problem occur in Uni3DETR since many more queries are used in inference?**
>
> A7: During training, queries regarding negative samples will be assigned as the background category through Hungarian matching, and will be learned towards a low objectness score in the classification layer. Then, in the inference process, the confidence score of most false negatives will be low. Therefore, such a false negative problem does not quite matter in our problem.

---

> > ### Comment · Area_Chair_qrjY · 2023-08-20
> >
> > Dear reviewer,
> >
> > Please look over the author response and the other reviews and update your opinion.  Please ask the authors if you have additional questions before the end of the discussion period.

---

### Author Rebuttal · Authors · 2023-08-08

Dear all reviewers

We sincerely thanks for valuable comments and suggestions. We first address the common concerns, followed by detailed responses to each reviewer separately. We hope our responses clarify existing concerns and make these points clear.

**Q: Comparison of computational complexity against the baselines in Table 3.**

A:
* In this paper, we mainly target a unified structure. To ensure that the detector can accommodate both indoor and outdoor detection, we have, to a certain extent, made sacrifices in terms of efficiency, in order to prioritize its unified ability. We list the comparison of both performance and efficiency in the following Table. We can observe that the computational budget compared with these baselines is not significant: the inference time (latency) is almost the same as UVTR and the FLOPS is only 1.16% more than UVTR. In addition, we obtain significantly better detection performance on both indoor and outdoor datasets. The indoor AP is 16.8% higher than UVTR, and the average of indoor and outdoor AP is also 8.8% higher than it. Compared with VoxelNeXt, one model that mainly focuses on reducing the FLOPS of 3D detectors, we achieve more than 40% indoor AP improvement and more than 25% average AP improvement. Meanwhile, the increase in training time is also negligible:  2d 10h v.s. 2d 7h of UVTR. This demonstrates that our model can obtain better detection performance on both indoor and outdoor datasets, with a relatively subtle increase in computational complexity.
* As the first work to build a unified architecture for both indoor and outdoor detection, we mainly focus on the performance level. From now on, there is still no work that can achieve a good performance on both indoor and outdoor datasets. Therefore, here our Uni3DETR mainly addresses the problem of achieving a good detection performance on both indoor and outdoor datasets with a unified architecture. For model efficiency, there are many other general approaches to address it. For example, we can adjust the number and ratio of sparse and dense convolution layers. We can leave this as one of our future work.

Table: The comparison of both performance and efficiency of our Uni3DETR against previous works on the indoor SUN RGB-D dataset and the outdoor nuScenes dataset. The metrics are AP25 for SUN RGB-D and mAP for nuScenes.
|method |performance | | | efficiency| | |
| :--: | :--: | :--: | :--: | :--: | :--: | :--: |
| |avg. |AP-indoor |AP-outdoor |latency (s) |params (M) |FLOPS (G)
|Centerpoint (CVPR 2021) |37.75 |18.9 |56.6 |0.32 |9.17 |121.10 |
|PillarNet (ECCV 2022) |44.00 |28.2 |59.8 |0.31 |12.55 |100.10 |
|VoxelNeXt (CVPR 2023) |39.30 |18.1 |60.5 |0.29 |7.12 |42.57 |
|UVTR (NeurIPS 2022) |55.55 |50.2 |60.9 |0.51 |26.12 |451.12 |
|ours |64.35 |67.0 |61.7 |0.52 |26.71 |458.74|

---

### Decision · Program_Chairs · 2023-09-21

**Decision:**

Accept (poster)

**Comment:**

This work proposes a unified 3D detection framework that can be used for both outdoor and indoor environments.

Overall, reviewers are positive on this work.  The AC agrees that this is a important step forward as most work in current 3D detection focuses on just either outdoor or indoor, and commonly even just a single dataset.  Reviewers also point out that the work is not as "unified" as it can be as separate models, with different parameters such as voxel-size need to be trained for indoor vs outdoor.  Still by presenting an unified framework with an initial set of experiments,  this work can encourage researchers to further investigate this direction.

Authors are encouraged to incorporate reviewer suggestions and clarifications provided during the author response period into their paper (or supplemental materials) including:
1. Incorporate additional experiments provided for the rebuttal (comparison of computational cost, cross-dataset performance, affect of voxel size and other model design choices, additional comparisons against prior work, etc).
2. Incorporate clarifications, and improve writing to make sure claims about unified architecture is precise and tone down claims of "strong universal ability".
3. Adding discussion of limitations of the experiments and clarify key differences when training models for indoor vs outdoors.
4. Adding discussion of recent work that investigates having single model for multiple datasets for 3D object detection:
- Uni3D: A Unified Baseline for Multi-dataset 3D Object Detection, Zhang et al. CVPR 2023